# Mechanistic investigation of human maturation of Okazaki fragments reveals slow kinetics

Vlad-Stefan Raducanu [1,2], Muhammad Tehseen[1,2], Amani Al-Amodi[1], Luay I. Joudeh [1], Alfredo De Biasio [1] ✉ & Samir M. Hamdan [1] ✉

The final steps of lagging strand synthesis induce maturation of Okazaki fragments via removal of the RNA primers and ligation. Iterative cycles between Polymerase δ (Polδ) and Flap endonuclease-1 (FEN1) remove the primer, with an intermediary nick structure generated for each cycle. Here, we show that human Polδ is inefficient in releasing the nick product from FEN1, resulting in non-processive and remarkably slow RNA removal. Ligase 1 (Lig1) can release the nick from FEN1 and actively drive the reaction toward ligation. These mechanisms are coordinated by PCNA, which encircles DNA, and dynamically recruits Polδ, FEN1, and Lig1 to compete for their substrates. Our findings call for investigating additional pathways that may accelerate RNA removal in human cells, such as RNA pre-removal by RNase Hs, which, as demonstrated herein, enhances the maturation rate ~10-fold. They also suggest that FEN1 may attenuate the various activities of Polδ during DNA repair and recombination.

The unidirectional synthesis by DNA polymerases and the chemical bidirectionality of DNA force the replisome to copy parental strands via two distinct modes[1–6]. The leading strand is continuously replicated, while lagging strand synthesis is discontinuous, via the formation of short Okazaki fragments (OFs), extending for ~200 nucleotides (nt)[7]. OF synthesis is initiated by the polymerase α (Polα)-primase complex, which generates a hybrid primer of 8–12 RNA and 10–20 DNA nucleotides[8,9]. Thereafter, RFC loads PCNA onto the primer-template (P/T) junction[10] for enhanced Polymerase δ (Polδ) processivity during primer extension. Maturation of Okazaki fragments (MOF) is initiated as Polδ•PCNA encounters the RNA primer on the preceding OF, performing limited strand displacement (SD) synthesis and giving rise to a single-stranded 5′flap structure[11,12] (Fig. 1A). Flap endonuclease-1 (FEN1) cleaves the 5′flap[13–15] and generates a nick product (NP) that can then be sealed by DNA Ligase 1 (Lig1) (Fig. 1A)[5,11,16,17].

PCNA encircles duplex DNA, coordinating the activities of Polδ, FEN1, and Lig1 during MOF. Cryo-EM structures of the human and yeast Polδ•PCNA complexes bound to P/T[18,19] as well as human Lig1•PCNA

bound to a NP[20] show Polδ and Lig1 occupying a single PCNA monomer, while the remaining two PCNA monomers are available to recruit additional proteins in characteristic toolbelt fashion. In fact, human FEN1•Polδ•PCNA[18] and FEN1•Lig1•PCNA[20] toolbelts were previously reported via cryo-EM. Further, these were shown to play a role in coupling SD and flap cleavage activities in yeast[15] as well as assisting ligation in humans[20], respectively. However, the detailed kinetics of toolbelt formation and specific substrate hand-off mechanisms during MOF remain largely unknown, particularly in human cells.

In yeast, the FEN1•Polδ•PCNA toolbelt rapidly removes RNA through iterative cycles of SD followed by 5′flap cleavage (Fig. 1A)[15]. Since the nick position shifts after each cycle, this mechanism is called nick translation (NT) (Fig. 1A). Maintaining RNA removal to 1 nt per cycle enhances the rate and processivity of the yeast NT reaction[15]. The rate of SD for the first 3 nt progressively decreases ~35-fold, as the flap length increases. In the presence of FEN1, the NT reaction predominantly cleaves 1-nt 5′flaps and consequently proceeds at rates that are comparable with the SD rate at the first nucleotide. Therefore,

[1]Bioscience Program, Division of Biological and Environmental Sciences and Engineering, King Abdullah University of Science and Technology, Thuwal 23955, Saudi Arabia. [2]These authors contributed equally: Vlad-Stefan Raducanu, Muhammad Tehseen. ✉e-mail: alfredo.debiasio@kaust.edu.sa; samir.hamdan@kaust.edu.sa

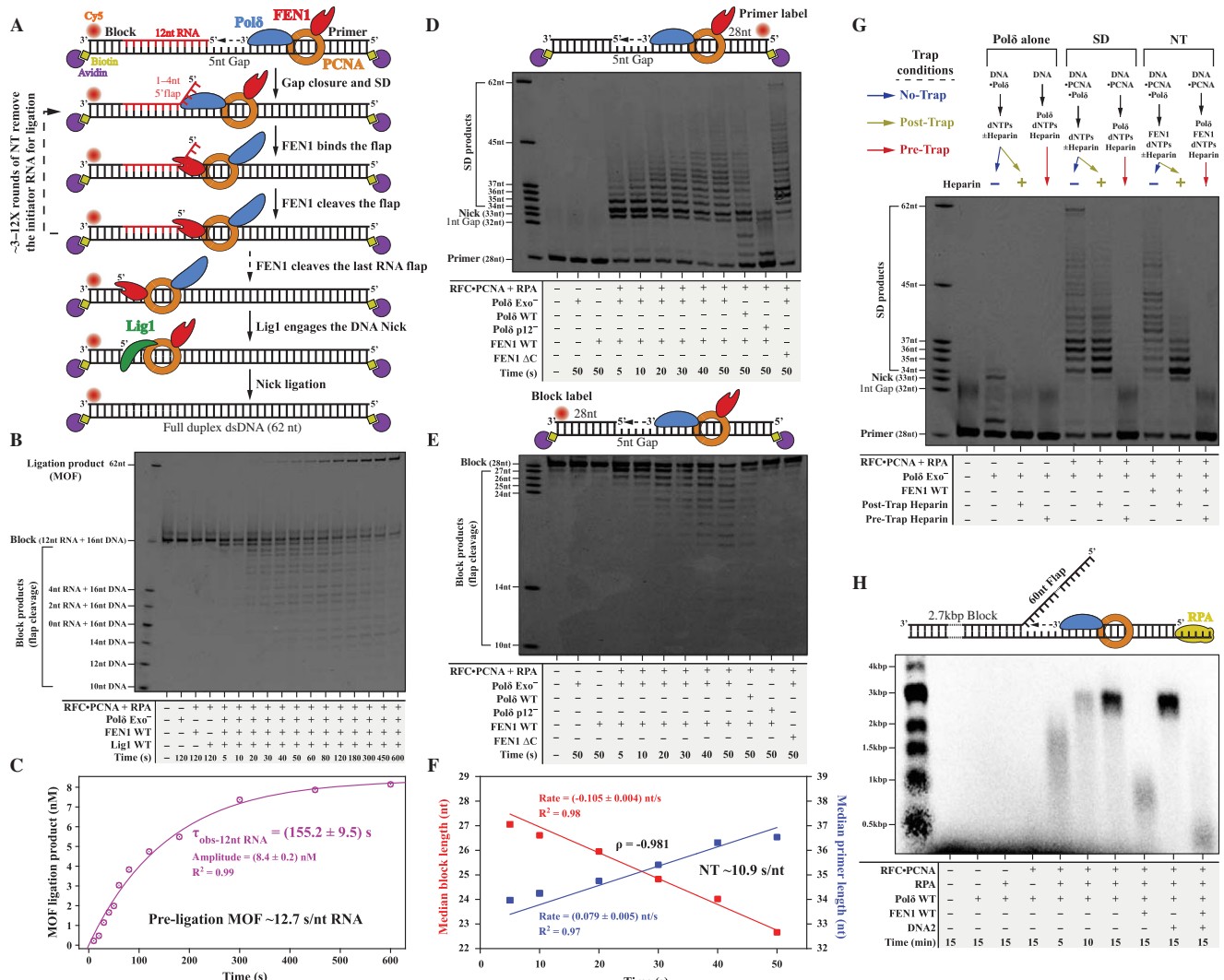

**Fig. 1 | Reconstitution of the human MOF reaction. A** Cartoon depiction of the maturation of Okazaki fragments (MOF) and nick translation (NT) reactions. **B** Reaction products for the reconstituted human MOF: 250 nM Polδ, 250 nM FEN1, 250 nM Lig1, and DNA Sub#1 (Supplementary Fig. 7) at room temperature (RT). **C** Quantification of the time dependence of the ligated MOF product yield from panel **B**. The experimental datapoints were fitted to a single-exponential product-formation burst equation [MOF Ligated Product = Amplitude $* (1 − e^{−t/\tau_{\mathrm{obs}−12nt\,RNA}})$]. The pre-ligation MOF processing time was calculated by subtracting 3.3 s (the time needed for the final ligation step; Supplementary Fig. 1B) from $\tau_{\mathrm{obs}−12nt\,RNA}$ (155.2 s) and then the result was divided by 12 nt (the length of the RNA region of the block). **D, E** NT products monitored through the extension of the primer oligonucleotide (**D**) or through the cleavage of the block oligonucleotide (**E**): 250 nM Polδ, 250 nM FEN1, RT, DNA Sub#2, and Sub#3 (Supplementary Fig. 7). **F** Quantification of the reaction rates presented in panels **D** and **E**. Median product lengths beyond the first

NP were determined for each reaction time as described in "Methods". The experimental datapoints were fitted to linear dependencies with fixed intercepts (28 nt for block length reduction and 33 nt for primer length increase). The NT processing time per nucleotide was calculated as the inverse of the average of the absolute values of the median primer increase and block reduction rates. **G** Processivities of the strand displacement (SD) and NT reactions in the presence of a Polδ trap competitor: 100 nM Polδ, 100 nM FEN1, 20 ng/μL heparin, and DNA Sub#2 for 30 s at 37 °C. **H** SD and NT activities on a long (2.7 kbp) substrate containing a long pre-formed flap (60 nt) at the end of a 30-nt gap. Polδ activity was monitored through the incorporation of radiolabeled deoxynucleotides: 30 nM Polδ WT, 30 nM FEN1, 600 nM RPA, 30 nM DNA2 at 37 °C. The long substrate and the radioactivity-based assay are detailed in the Methods section. Source data are provided as a Source Data file.

FEN1 enhances the rate of NT relative to SD by keeping the flap length short. Additionally, since increasing the flap length decreases SD processivity, keeping flaps short will enhance the processivity of the NT reaction[15].

A number of structural[18,19] and biochemical[21–23] studies suggest a considerably lower stability of human Polδ•PCNA on DNA when compared to yeast, suggesting that SD and NT during MOF may differ between the two systems. Although the structures of Polδ•PCNA complexes with P/T junctions are similar between yeast and humans, yeast Polδ makes ~50% more contacts with PCNA[19]. In fact, the lifetime of Polδ•PCNA on a P/T junction in yeast is ~50-fold greater than in humans[21], and its processivity does not limit OF synthesis[21,22], while up

to ~30% of human Polδ•PCNA complexes may dissociate before even finishing an OF[23]. Under SD synthesis conditions, several additional factors act against polymerase advance into the duplex DNA, namely, polymerase idling[11,15], the growing 5′flap acting as a molecular brake[15], and the transient binding of RPA[12]. Therefore, it can be envisioned that the lower stability of the human Polδ•PCNA complex on DNA can lead to even more dramatic differences in SD and NT during MOF.

Herein, we reconstitute human MOF and employ biochemistry, single-molecule imaging, and a variety of bulk-fluorescence assays to obtain a comprehensive understanding of MOF kinetics and substrate hand-off mechanisms. Lower stability of human Polδ•PCNA complexes on DNA results in a MOF reaction that is more dynamic, less

processive, and remarkably slower than in yeast. We also show that additional pathways such as RNA pre-removal by RNase H2 might be critical to accelerate MOF in humans as opposed to the intrinsically fast and efficient NT pathway in yeast. In addition, we address the implications of our findings in correcting the potential errors introduced by the proofreading-deficient Polα. Finally, we also discuss the consequences of our results as to how FEN1 may attenuate the various activities of Polδ during DNA repair and recombination.

## Results

### Reconstitution of the human MOF reaction

MOF was reconstituted on a substrate containing a 5-nt gap between the nascent and previous OFs. The previous OF (termed block) contains 12-nt RNA at the 5′ end and a Cy5 label at the 3′ end for monitoring the NT reaction (Fig. 1A). Bilateral terminal Biotin–Avidin blocking of the DNA free ends was included to prevent PCNA sliding off the substrate[15,24]. Protein concentrations were kept above their dissociation constants (see below) to saturate the formation of any intermediary complexes. The NT reaction generated a time-dependent series of block cleavage products whose ligation started after a delay interval of ~20 s (Fig. 1B). The ligation products were quantified as a function of time (Fig. 1C) and yielded an apparent MOF of ~155 s. Since the NT reaction did not proceed considerably beyond the 12-nt RNA (Fig. 1B), the pre-ligation processing time for RNA removal was estimated to be ~13 s/nt (Fig. 1C). In a control experiment, we showed that Lig1 sealed a nick within ~3 s and discriminated against a nick containing 1-nt RNA (Supplementary Fig. 1A, B)[25]. Therefore, RNA removal by NT is the rate-limiting step during MOF.

To better understand the mechanism underlying the NT reaction, we monitored NT primer extension (Cy5-labeled primer; Fig. 1D) and 5′ flap cleavage (Cy5-labeled block; Fig. 1E) in the absence of Lig1. The median length of the primer (Fig. 1D) and the block (Fig. 1E) exhibited a strong anti-correlation ($\rho = -0.981$), with median rates of primer increase and block reduction at ~0.08 nt/s and ~0.10 nt/s, respectively, thus yielding a median human NT processing time of ~11 s/nt (Fig. 1F). This directly-determined NT processing time is in agreement with the pre-ligation MOF processing time determined above (~13 s/nt; Fig. 1C). The coupling between SD and flap cleavage requires the interaction of FEN1 with PCNA, since a C-terminal-truncated FEN1 (FEN1 ΔC) that cannot interact with PCNA but retains endonuclease activity[26] (Supplementary Fig. 1U), was incapable of competing with Polδ during NT (Fig. 1D, E and Supplementary Fig. 1C, D).

We then investigated the processivity of Polδ during SD and NT reactions by including a heparin DNA-competitor that traps DNA-unbound Polδ[15,18,24] (Fig. 1G). For both reactions, the trap did not affect gap closure efficiency, yet considerably reducing SD processivity to 1–4 nt (80% of total products). The limited SD processivity of human Polδ was further confirmed by challenging preassembled active Polδ with catalytically inactive Polδ (Supplementary Fig. 1T). Under these conditions, the SD median product lengths could decrease by >4-fold and the products patterns were nearly identical to those produced by the heparin trapping experiment. In the presence of FEN1, the processivity was also 1–4 nt, with a pattern toward lower processivity (Fig. 1G). These data show that human NT is un-processive and can translate the nick by a maximum of 4 nt.

Remarkably, yeast NT is ~55-fold faster than in humans and can processively consume the entire 28-nt block[15]. A series of control experiments excluded the possibility that this difference in NT may result from using exonuclease activity-deficient Polδ[15,27] (Polδ Exo⁻) (Fig. 1D, E and Supplementary Fig. 1C, D) or the presence of the extra p12 subunit in humans (Fig. 1D, E and Supplementary Fig. 1C, D). In SD, Polδ WT was ~3-fold slower than Polδ Exo⁻ (Supplementary Fig. 1P, R versus Supplementary Fig. 1K, N) and generated more pre-initial-NP products (Supplementary Fig. 1S), due to idling at the first NP[11,15] and additional 3′-to-5′ exonuclease activity of Polδ WT. In NT however, Polδ

WT and Polδ Exo⁻ produced nearly identical rates beyond the first NP (Supplementary Fig. 1Q, R versus Supplementary Fig. 1L, N). We also showed that protein concentrations saturated in our NT reaction, since the rate of NT was unaffected at 10-fold lower protein concentrations (Supplementary Fig. 1F, G versus Fig. 1D–F). Interestingly, SD activity for the first 1–4 nt in 5 s is highly similar in human and yeast (Supplementary Fig. 1E, K)[15], suggestive of similar SD rates. In addition, yeast and human FEN1 display similar kinetics[15,28], and we showed that the cleavage rate on short flaps is relatively independent of 5′flap length and much faster than the NT rate (Supplementary Fig. 1U).

These findings point to communication between Polδ and FEN1 as the reason for the different kinetics of NT between human and yeast. In fact, in human FEN1 slightly inhibited the rate and processivity of NT compared to SD, while in yeast FEN1 dramatically increased them[15]. This was evident based on the persistence of NPs upon cleavage of the first 1–4 nt, as opposed to their gradual progression in yeast (Supplementary Fig. 1E, F, H)[15], suggesting that NPs are not being transferred efficiently from human FEN1 to Polδ. In support of this conclusion, increasing the reaction temperature to 37 °C selectively increased the rate of SD ~3.5-fold (Supplementary Fig. 1K, N versus Supplementary Fig. 1E, G) without affecting the rate of NT (Fig. 1D–F and Supplementary Fig. 1F, G versus Supplementary Fig. 1L–O). The stimulation of SD activity at 37 °C was likely aided by the enhanced thermal melting of the dsDNA in a sequence-dependent manner (Supplementary Fig. 1I, J)[15].

Finally, we showed that SD restart inhibition by FEN1 occurs under conditions of excessive Polδ-mediated SD synthesis on a long substrate in the presence of high RPA concentration and a long pre-formed 5′flap (Fig. 1H). Under these conditions, the molecular break imposed by a short nascent 5′flap was removed, allowing Polδ to mediate multiple cycles of SD synthesis. However, since FEN1 cleavage is inhibited on long RPA-coated 5′flaps[14], it is possible that part of this inhibition is due to a repeated competition between FEN1 and Polδ for PCNA binding, especially since FEN1 can bind to PCNA with higher stoichiometry[29] than Polδ[18]. To address this, we included the DNA2 helicase–nuclease that shortens the 5′flap and removes RPA for proper substrate engagement by FEN1[30–35], and showed that SD restart inhibition by FEN1 increased dramatically as observed on the short substrates.

### Lig1, but not Polδ, can efficiently release the NP from FEN1

We then focused on substrate conformational requirements and hand-off mechanisms among MOF proteins. We previously deciphered conformational states during the FEN1 catalytic cycle using single-molecule FRET[13,14,36,37] (Fig. 2A). The first state is a linear flap substrate prior to FEN1 binding, the second is a bent FEN1-bound flap substrate, the third is a bent FEN1-bound NP, and the last is a released linear NP. In the internal-labeling scheme, the donor and acceptor are located on the duplex arms of a linear flap substrate (Fig. 2A)[13,14]. Upon FEN1 binding and bending, the distance between the two fluorophores decreased, and, therefore, FRET increased for the second and third conformational states. Following FEN1 dissociation, FRET of the extended NP decreased to an efficiency that is slightly lower than that of the initial linear flap substrate[14]. In the flap labeling scheme, the donor and acceptor are located on the 3′ duplex region and at the tip of the 5′flap, respectively (Supplementary Fig. 2A). The substrate started with a high FRET state, and, upon binding and bending, the distance between the fluorophores increased to result in a lower FRET state. Upon flap cleavage, the acceptor signal and, consequently, FRET were lost. Thus, the transition between the last two states could no longer be detected. For both labeling schemes, the bent states could be accessed without progressing to flap cleavage by replacing FEN1 WT with the catalytically inactive mutant FEN1 D181A[13], even in the presence of Mg²⁺.

Using the internal-labeling scheme, we started by testing the interaction of individual MOF proteins with the flap substrate in the presence or absence of RFC•PCNA (Fig. 2B). Polδ increased FRET only

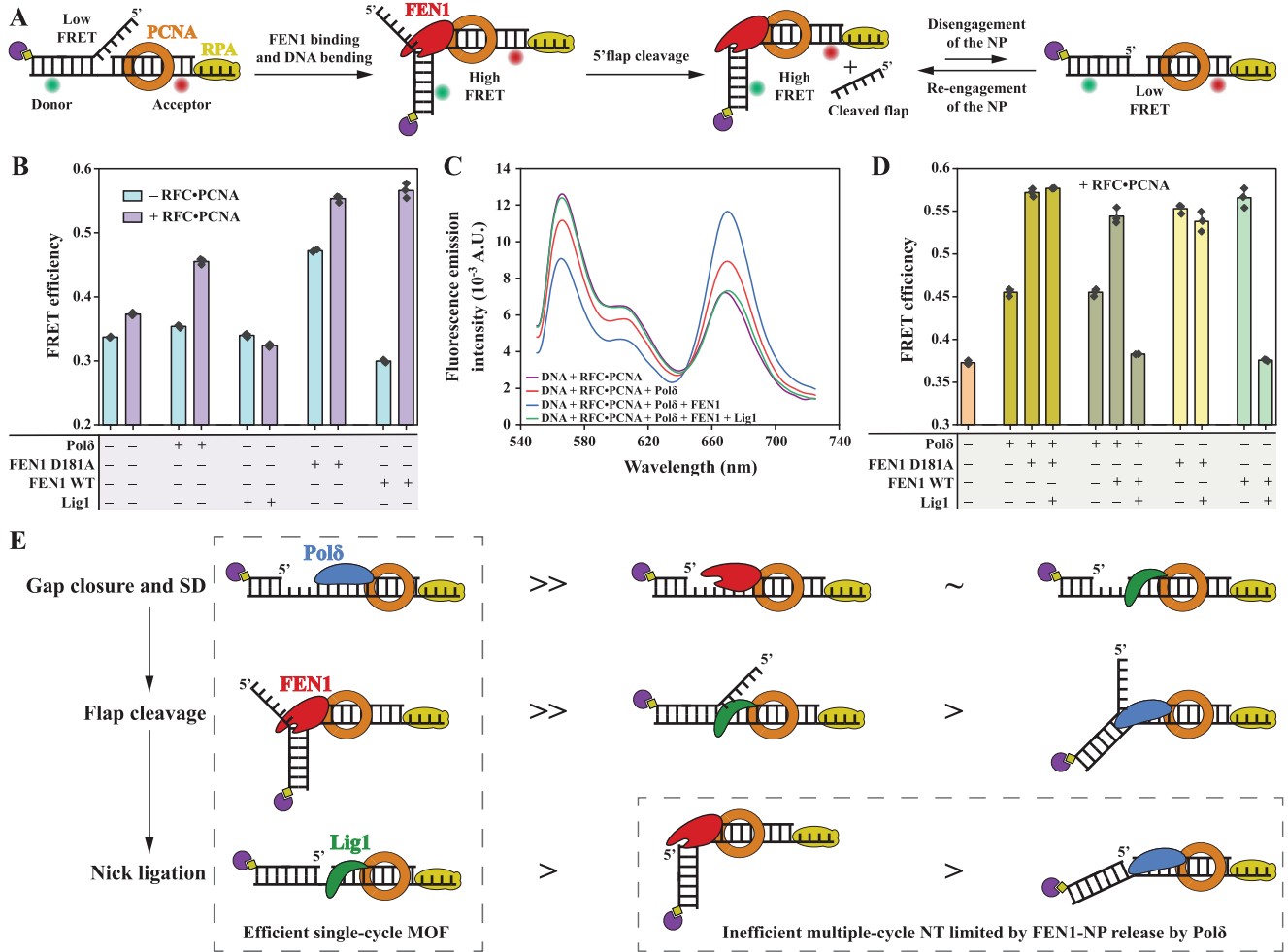

**Fig. 2 | Substrate hand-offs among MOF proteins. A** Cartoon representation of the internal-labeling scheme used to monitor flap and NP engagement. **B** Apparent FRET efficiencies of the double flap (Sub#12; Supplementary Fig. 7) upon addition of individual MOF proteins in the presence and absence of RFC•PCNA. The bar chart illustrates the mean (as bar height) and one standard deviation (as error bar) of three independent measurements. **C** Examples of emission spectra of the internally labeled double flap (Sub#12) upon addition of various combinations of MOF proteins. **D** Apparent FRET efficiencies of the internally labeled double flap (Sub#12)

upon the addition of various combinations of MOF proteins in the presence of RFC•PCNA. The bar chart illustrates the mean (as bar height) and one standard deviation (as error bar) of three independent measurements. Conditions for panels **B**–**D**: 15 nM FEN1, 250 nM Polδ, 500 nM Lig1. **E** Cartoon representations of the various affinities of MOF proteins to intermediate DNA structures. For each intermediate DNA substrate, the affinity is decreasing from left to right. » denotes much higher affinity, > denotes higher affinity, and - denotes similar affinity. Source data are provided as a Source Data file.

in the presence of RFC•PCNA, which we attributed to flap substrate bending, while Lig1 did not induce any considerable FRET change under both conditions. FEN1 WT in the absence of PCNA cleaved the flap and released the linear low FRET NP. This is expected since FEN1 rapidly cleaves the 5′flap[13] and cannot efficiently rebind and bend the NP at 100 mM KCl ($K_D > 580$ nM[14]). Remarkably, in the presence of RFC•PCNA, FEN1 WT rebound the NP and maintained it in a bent state of similar FRET to the FEN1 D181A-bound bent flap substrate.

With the establishment of FRET states of individual MOF proteins, we next investigated the combined sequential action of these proteins in the presence of RFC•PCNA (Fig. 2C). Polδ and Lig1 were used at ~17- and 33-fold in excess of FEN1, respectively, to confer them a clear binding advantage. Both FEN1 WT and D181A were able to take over the DNA substrate in the presence of Polδ (Fig. 2D), demonstrating that FEN1 is superior in binding the flap and NP over Polδ. Lig1 was not able to release the flap substrate from FEN1 D181A but could release the NP from FEN1 WT, irrespective of Polδ presence (Fig. 2D). In a control experiment, we used the flap labeling scheme (Supplementary Fig. 2A) to show that addition of Polδ and/or Lig1 does not impair FEN1 cleavage (Supplementary Fig. 2B, C). While Lig1 is expected to compete with Polδ on NPs, we found that it can also

destabilize Polδ on flap substrates (Supplementary Fig. 2D). In fact, Lig1 decreased the median size of SD products by ~8 nt (Supplementary Figs. 2E and S2G, top), and this required the interaction of its N-terminus with PCNA (Supplementary Fig. 2E and S2G, top). However, in MOF, the ligation yield was not affected by replacing Lig1 WT with Lig1 ΔN (Supplementary Fig. 2F, G, bottom), but Lig1 WT was slightly more efficient in stopping the NT reaction earlier (Supplementary Fig. 2F). It is possible that the slow rate of NT in humans combined with efficient release of the NP from FEN1 by Lig1 eliminated some of the Lig1 dependence on PCNA. This is in contrast with the MOF reaction in archaea[38], and, presumably, in yeast, where removal of the Lig1-PCNA interaction resulted in a complete lack of MOF ligation products.

Collectively, these results, enabled us to sort the affinities for PCNA-loaded DNA substrate intermediates during MOF, without exact quantification (Fig. 2E). Gap closure by Polδ•PCNA is largely unaffected by the presence of FEN1 (Supplementary Fig. 1E, K versus 1F, L; Supplementary Fig. 1S) or Lig1 (Supplementary Fig. 2E). SD activity of Polδ•PCNA generates a flap structure which is efficiently won by FEN1 despite the presence of excess Polδ or Lig1 (Fig. 2D). FEN1 wining the double flap over Polδ is well-supported since FEN1 binds this substrate

with high affinity ($K_D$ < 5 nM[13,14]), while Polδ binds it with >10-fold lower affinity (see below). A flap substrate can also be won by Lig1 over Polδ (Supplementary Fig. 2D), which should subsequently be won by FEN1. Last, the NP is efficiently engaged by Lig1 in the presence of FEN1, irrespective of Polδ (Fig. 2D). Most surprisingly, the NP is won by FEN1 over Polδ despite a large excess of Polδ, demonstrating that Polδ is inefficient in releasing the NP from FEN1 to restart another cycle of SD.

### Communication between FEN1 and Polδ at the single-molecule level

We then employed single-molecule imaging to capture substrate hand-off between FEN1 and Polδ in real-time. We started by investigating the effect of PCNA on FEN1 kinetics. Using the flap labeling scheme (Supplementary Fig. 3A) to determine DNA bending (decrease in FRET from 0.8 to 0.5) and 5′flap cleavage (departure of acceptor) (Fig. 3A), we obtained the dwell time of the bent substrate before 5′flap departure (~165 ms) (Fig. 3B). The different time regimes of 5′flap departure versus acceptor photobleaching allows for a clear differentiation between the two processes[13]; in the presence of FEN1 WT 5′flap departure occurs within <1 s (Fig. 3A–C), while in the presence of FEN1 D181A the acceptor signal in the bent conformer persists for tens of seconds without photobleaching (Supplementary Fig. 3A). As 5′flap departure is spontaneous[13,15,39], the bent conformer dwell time directly reflects the single turnover cleavage kinetics[13,14]. In addition, the presence of PCNA had only a minor effect on these, with the dwell time increasing to ~210 ms (Fig. 3C). This minor increase excludes the possibility that PCNA reduces FEN1 single turnover cleavage activity during NT. To follow the NP, we employed the internal-labeling scheme (Supplementary Fig. 3B) for determining substrate bending (increase in FRET from 0.3 to 0.5) and NP fate, up to the unbending step (decrease in FRET to 0.25) (Fig. 3D and Supplementary Fig. 3C)[13,14]. In the absence of PCNA, FEN1 maintained the bent state for ~310 ms (Supplementary Fig. 3D). In its presence, FEN1 bent the substrate to a slightly higher FRET value (~0.6; probably due to a PIFE effect on the acceptor[36,40]), but never returned or passed through an unbent state (Fig. 3E). Since cleavage is not impaired by PCNA (Fig. 3B, C and Supplementary Fig. S2B, C), this long-lived bent state must represent FEN1 binding to its NP, at least after the initial ~210 ms cleavage time (Fig. 3C).

We next used the internal-labeling scheme to monitor the handover of the flap substrate from Polδ to FEN1. Polδ increased FRET from 0.3 to 0.45 only in the presence of trapped PCNA (Fig. 3F, G) which we attributed to assembly of the Polδ•PCNA complex and bending of the DNA substrate. However, the assembly yield was only ~70% (Supplementary Fig. 3E), which is consistent with previous reports[23] and could not be empirically improved by varying the experimental conditions. To observe the hand-off from Polδ to FEN1, Polδ•PCNA was prebound to the flap substrate in the absence of dNTPs (0.45 FRET state) (Fig. 3I). Upon injection of FEN1, an abrupt increase in FRET to >0.55 was observed (Fig. 3I). This transition to higher FRET was due to FEN1•PCNA bending the flap substrate and, subsequently, the NP upon 5′flap cleavage, as it is similar to the FRET states upon flap or NP bending by FEN1•PCNA (Fig. 3E and Supplementary Fig. 3F). Interestingly, we consistently observed a transition of FRET states from 0.45 to >0.55 (Supplementary Fig. 3F), with no unbent intermediate (Fig. 3I). This indicates that Polδ might hand-off an already bent flap substrate to FEN1, as previously suggested[18]. However, it remains possible that a transition to an unbent intermediate within our 50 ms temporal resolution may be masked by the near diffusion-limited FEN1 binding and bending of the flap substrate[13,14]. The observation that FEN1•PCNA maintained the NP in the bent state (Fig. 3H and the FRET state after transition in Fig. 3E versus the FRET state after transition in Fig. 3I; Supplementary Fig. 3F) provides clear evidence that Polδ is inefficient in timely releasing the NP from FEN1•PCNA.

### Toolbelt formation is not limiting for MOF

Next, we focused on quantifying the competition among MOF proteins for PCNA. FEN1 interacts with PCNA in solution with moderate affinity ($K_D$ ~ 70 nM[41]). EMSA revealed that Polδ and Lig1 formed complexes with PCNA in solution with $K_D$ values of ~15 and ~30 nM, respectively (Supplementary Fig. 4A–D). Therefore, the tighter binding of Polδ to PCNA in solution cannot explain why Polδ is not able to compete with FEN1 on the NP. In a series of experiments, we demonstrated that although Polδ is inefficient in displacing FEN1 from the NP (Figs. 2D and 3I), it can still bind PCNA to form a toolbelt with FEN1 on the NP (Fig. 4A). In these experiments, we prebound PCNA on a double flap DNA-containing Cy3 and monitored FEN1 binding and 5′flap cleavage via protein-induced fluorescence quenching (PIFQ) of Cy3[36] (Fig. 4A). In this assay, the interactions between the Cy3 placed in the vicinity of the double flap junction and the 5′flap restrict the fluorophore photoisomerization, which in turn creates a hyper-fluorescence state[36]. FEN1 disrupts these interactions by bending the DNA and threading the 5′flap though a capped helical gateway and further by cleaving the 5′flap[13,42]. While additional interactions might be created between FEN1 residues and Cy3, their overall strength is lower than those between the 5′flap and Cy3 in the initial state before FEN1 binding, leading to an overall higher Cy3 photoisomerization; this results in fluorescence quenching via PIFQ[36] upon FEN1 binding and 5′flap cleavage whereafter a NP was generated. Upon Polδ addition, a subsequent quenching effect by Polδ's iron–sulfur cluster[43,44] (termed FeSQ) was observed (Fig. 4A), which we attributed to Polδ binding to PCNA. To quantify Polδ binding to the FEN1•PCNA•NP complex, we replaced the double flap DNA with a pre-formed NP and we used FEN1 D181A to suppress FEN1 exonuclease activity (Fig. 4B). Increasing Polδ concentration resulted in a greater amplitude of quenching by FeSQ (Fig. 4B). The experimental datapoints were fitted to a quadratic dependence, yielding an apparent dissociation constant of ~4.5 nM (Fig. 4B). In the presence of an acceptor at the tip of the 5′flap, flap cleavage and Polδ association could also be visualized simultaneously via FRET, PIFQ, and FeSQ (Supplementary Fig. 4E).

We then validated the ability of FEN1 to bind PCNA and form a toolbelt with Lig1 on the NP. Lig1 was prebound to a PCNA-loaded NP that does not support ligation by incorporating a 3′ dideoxyC (ddC). In a control experiment, we verified that even a 4-fold molar excess of FEN1 could not release the NP from Lig1 (Supplementary Fig. 4F). To measure FEN1 binding to Lig1•PCNA•NP, an acceptor was placed on the NP for donor-labeled FEN1 titration (Fig. 4C). Since donor emission increased linearly with FEN1-Cy3 concentration, we could not rely on apparent FRET efficiency for quantification of the results. Instead, the total emission spectrum at each FEN1 concentration was fitted to a linear combination of Cy3 and Alexa647 individual spectra. The direct excitation of Alexa647 in the absence of FEN1-Cy3 was subtracted from deconvoluted Alexa647 emission at each FEN1-Cy3 concentration. The FRET-stimulated increase in Alexa647 emission was plotted against FEN1-Cy3 concentration and fitted to a quadratic dependence, yielding an apparent affinity of ~20 nM (Fig. 4D).

It should be noted that the affinity constants presented in Fig. 4B and D represent apparent values, mainly due to two assumptions. First, we assume that FEN1 is stably bound into the FEN1•PCNA•NP complex (Fig. 4B) and that Lig1 is stably bound into the Lig1•PCNA•NP complex (Fig. 4D), during the titration of the partner proteins. FEN1 binds the PCNA-loaded NP with ~5 nM affinity (see below), while Lig1 binds the NP with ~3 nM affinity in the absence of PCNA[45] and probably with even stronger affinity in the presence of PCNA. Therefore, under our experimental conditions, both FEN1 and Lig1 should be bound in a proportion of >98%. The second assumption is that the excess DNA-unbound pre-complexed proteins do not create significant kinetic inhibition for the binding of their toolbelt partners. Nevertheless, such a binding kinetic inhibition can only result in weaker apparent affinities. Therefore, the toolbelt formation affinities might be even stronger than

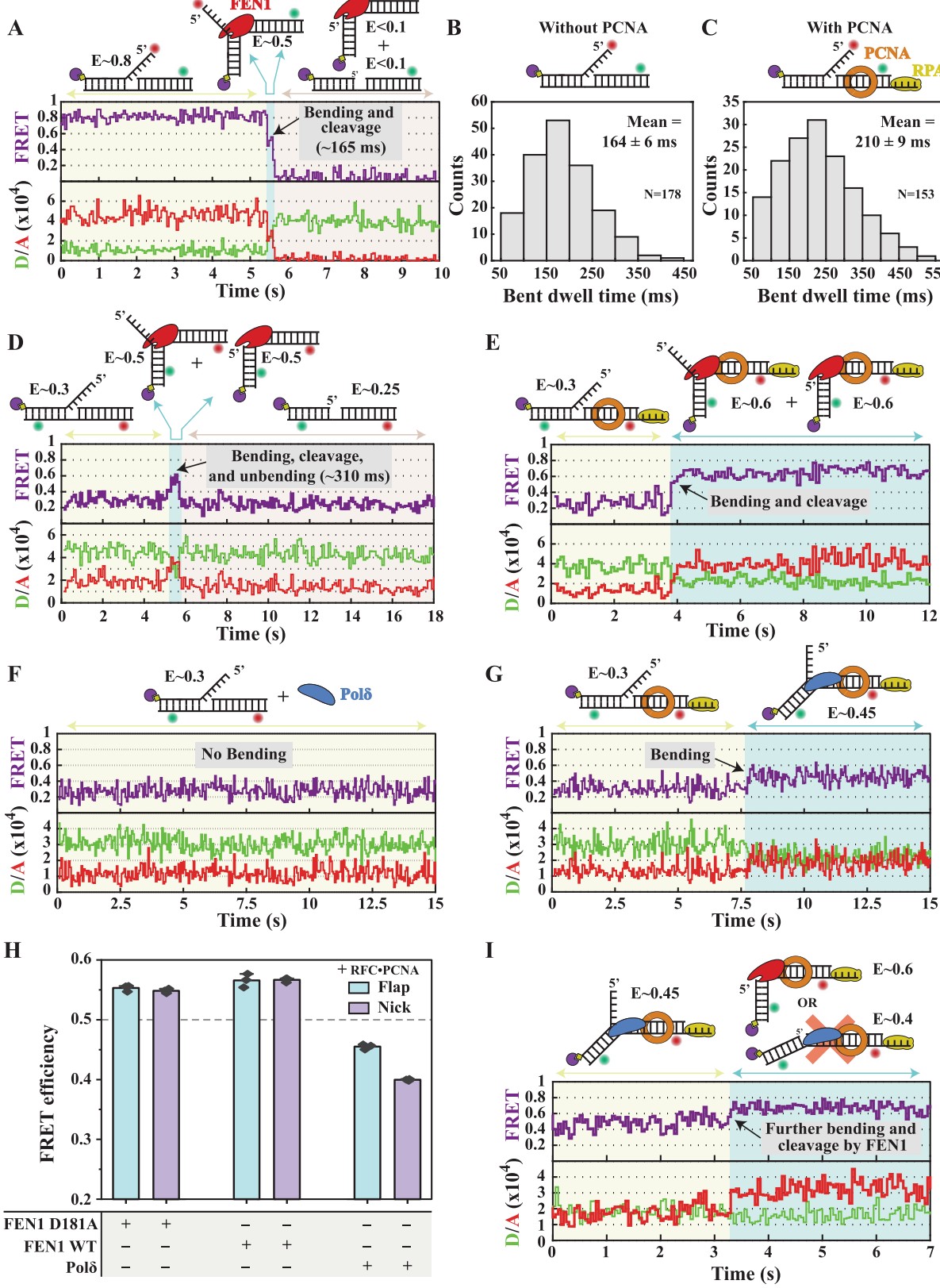

presented, thus further reinforcing the conclusion that toolbelt formation is not limiting.

**Kinetic competition for substrates regulates the toolbelts**

The fact that toolbelt formation through PCNA association is not limiting for MOF indicates that competition among Polδ, FEN1, and Lig1

for DNA must be the key regulator of MOF kinetics. We measured the dissociation constant of FEN1 on its NP in the presence of PCNA via bulk FRET titration (Fig. 5A). FEN1 D181A was used to prevent FEN1 exonuclease activity. The dissociation constant was ~5.3 nM (Fig. 5B), >100-fold lower than in the absence of PCNA ($K_D$ > 580 nM[14]). Next, we estimated the time required for FEN1 to dissociate from DNA through a

**Fig. 3 | Monitoring substrate hand-offs between FEN1 and Polδ at the single-molecule level. A** Representative single-molecule cleavage time trace of the double flap (Sub#23; Supplementary Fig. 7) by FEN1 (250 nM) through the flap labeling scheme at 50 ms temporal resolution. **B, C** Histograms of the distributions of dwell times of the bent conformer before cleavage from the flap labeling scheme in the (**B**) absence (Sub#23) and (**C**) presence (Sub#22; Supplementary Fig. 7) of preloaded PCNA. The indicated mean and error of the dwell time distributions represent the raw arithmetic mean and standard error of the mean of the raw datapoints that were binned into the histograms, without additional histogram fitting. Representative single-molecule cleavage time trace of the double flap by FEN1 (250 nM) through the internal-labeling scheme in the (**D**) absence (Sub#13; Supplementary Fig. 7) and (**E**) presence (Sub#12; Supplementary Fig. 7) of preloaded PCNA at 100 ms temporal resolution. Flap cleavage must have occurred in

<3 frames. Representative single-molecule bending time trace of the double flap by Polδ (250 nM) through the internal-labeling scheme in the (**F**) absence (Sub#13) and **G** presence (Sub#12) of pre-loaded PCNA at 50 ms temporal resolution. **H** Apparent FRET efficiencies of the internally labeled double flap and NP (Sub#12 and Sub#24; Supplementary Fig. 7) upon addition of FEN1 D181A, FEN1 WT, or Polδ (250 nM each) in the presence of RFC•PCNA determined from bulk steady-state fluorescence measurements. The bar chart illustrates the mean (as bar height) and one standard deviation (as error bar) of three independent measurements. **I** Representative single-molecule time trace showing the hand-off of the double flap (Sub#12) from Polδ (250 nM) to FEN1 (250 nM) in the presence of pre-loaded PCNA at 50-ms temporal resolution. Flap cleavage must have occurred in <5 frames after FEN1 engagement. Source data are provided as a Source Data file.

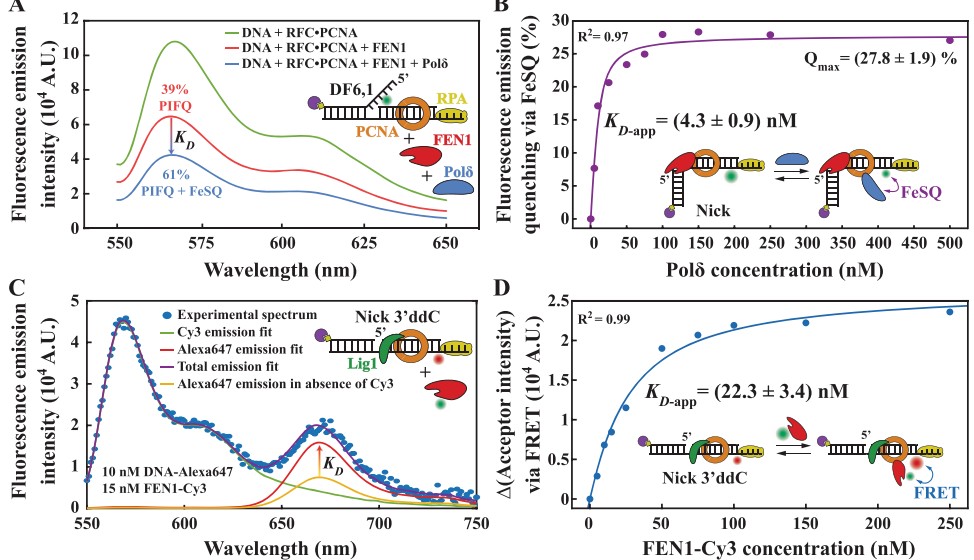

**Fig. 4 | Formation of the FEN1•Polδ•PCNA and FEN1•Lig1•PCNA toolbelts.**
**A** Emission spectra of a Cy3-labeled flap substrate (Sub#27; Supplementary Fig. 7) in the presence of FEN1 WT (15 nM) and Polδ (250 nM). FEN1 binding results in protein-induced fluorescence quenching of Cy3 (PIFQ), and Polδ binding results in additional quenching by its iron–sulfur cluster (FeSQ). **B** Polδ binding curve based on the experiment presented in panel A. FEN1 D181A (250 nM) was prebound to PCNA-loaded NP (Sub#28; Supplementary Fig. 7), and this complex constitutes the substrate for additional Polδ titration. The experimental datapoints were fitted to a dependence proportional to Eq. (1) ("Methods"). **C** Emission spectra of PCNA-loaded Lig1 (500 nM)-bound Alexa647-labeled nick DNA (Sub#30; Supplementary

Fig. 7) in the presence of Cy3-labeled FEN1 (15 nM). The nick DNA substrate contained a 3′ ddC to prevent ligation. Experimental spectra datapoints were fitted with a linear combination of Cy3 and Alexa647 spectra. **D** Quantification of the data presented in panel **C**. For various FEN1-Cy3 concentrations, the total emission spectra were fitted to linear combinations of Cy3 and Alexa647 spectra. The increase of the coefficient for the Alexa647 part of the linear spectral combination (via FRET) was recorded and plotted against its corresponding FEN1-Cy3 concentration. The experimental datapoints were fitted to a dependence proportional to Eq. (1) ("Methods"). Source data are provided as a Source Data file.

---

bulk-fluorescence recovery experiment[28]. FEN1 was prebound to its NP, generating a high FRET state, under competition with excess uncleavable flap substrate to capture any dissociated FEN1 (Fig. 5A, B). FEN1 dissociated from the NP within ~1.8 s (Fig. 5C, top), with a dissociation rate $k_{off\text{-}FEN1\text{-}NP} = 0.56\,s^{-1}$. The association rate can be estimated as $k_{on\text{-}FEN1\text{-}NP} = 1.1 \times 10^8\,M^{-1}s^{-1}$, which is at the diffusion limit.

Next, we verified whether Lig1 released the NP from FEN1 via an active or passive mechanism. FEN1 was prebound to the NP as described above, yet this time being released via the addition of a large excess of Lig1. In this case, FEN1 dissociated within ~0.9 s (Fig. 5C, bottom). This ~2-fold faster dissociation compared to the DNA trap condition indicates that Lig1 actively releases FEN1 from its NP. The fluorescence recovery amplitude is similar in the case of Lig1 release and DNA trap capture (Fig. 5C), indicating that the NP is efficiently kept and probably sealed by Lig1 following FEN1 release.

To assess Polδ binding kinetics for a variety of substrates, we turned our attention again to FeSQ[43,44]. Upon titrating Polδ to a PCNA-loaded Cy3-labeled NP, a gradual decrease in Cy3 emission intensity was observed (Fig. 5D). The affinity of Polδ to the 2-nt gap was estimated at ~5.4 nM (Fig. 5E), which is consistent with a previous report[46].

As the gap turned into a nick and then a 5′flap, the affinity decreased ~5-and ~10-fold, respectively (Fig. 5E). These data suggest that, from a kinetics perspective, Polδ senses the 5′block as it enters SD mode. Using bulk-fluorescence recovery between the Polδ's iron–sulfur cluster and Cy3 on the DNA (FeSQ elimination), we estimated Polδ dissociation times from the 2-nt gap and NP at ~3.5 s and ~1.1 s, respectively (Fig. 5F). The dissociation rate for the 2-nt gap is only slightly faster than the previously reported value for a P/T junction (~6.3 s)[23]. Based on dissociation constant and rate ($k_{off\text{-}Pol\delta\text{-}NP} = 0.9\,s^{-1}$), we obtained the association rate of Polδ to PCNA-loaded NP as $k_{on\text{-}Pol\delta\text{-}NP} = 3.5 \times 10^7\,M^{-1}s^{-1}$.

Taken together, kinetic parameters suggest that the probability of Polδ to win the competition for the NP over FEN1 can be estimated at one in four attempts of DNA engagement based on approximate association rates (Fig. 5G). The other three attempts are won by FEN1 and are mostly unproductive, since DNA is already NP, simply resulting in a delayed dissociation, even in the exonuclease mode ($k_{exo\text{-}FEN1\text{-}5nt\ Gap} = 0.01\,s^{-1} < k_{exo\text{-}FEN1\text{-}NP} = 0.02\,s^{-1} \ll k_{off\text{-}FEN1\text{-}NP} = 0.56\,s^{-1}$; Fig. 5C versus Supplementary Fig. 5A–D). Without a strong active hand-off to Polδ (Figs. 2D and 3I), the three unproductive FEN1

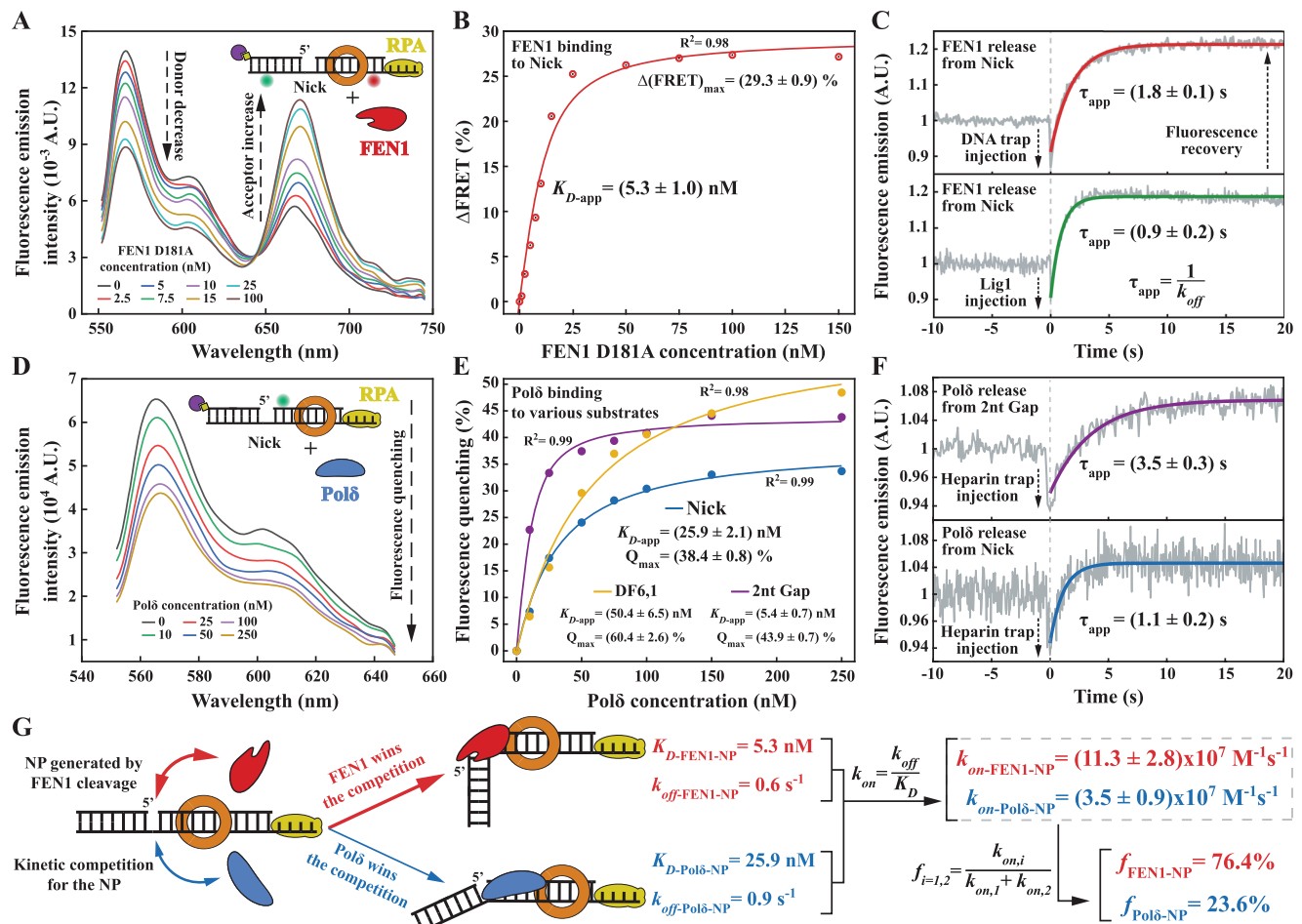

**Fig. 5 | Kinetic competition between FEN1 and Polδ for the NP. A** Fluorescence emission spectra of internal labeled NP (Sub#24; Supplementary Fig. 7) in the presence of pre-loaded PCNA at various concentrations of FEN1 D181A. **B** Quantification of the data presented in panel A. The experimental datapoints were fitted to a dependence proportional to Eq. (1) ("Methods"). **C** Fluorescence recovery of the Cy3 donor presented in panel A upon addition of a large excess of FEN1 DNA trap (top; 5 μM of unlabeled phosphotiolated nonequilibrating double flap; Sub#31; Supplementary Fig. 7) or Lig1 (bottom; 2 μM). The experimental datapoints were fitted to Eq. (2) ("Methods"). **D** Fluorescence emission spectra of Cy3-labeled NP (Sub#28; Supplementary Fig. 7) in the presence of pre-loaded PCNA at various concentrations of Polδ. **E** Quantification of the data presented in panel **D** and of similar experiments in which the NP was replaced with either a 2-nt gap

(Sub#29; Supplementary Fig. 7) or a double flap (Sub#27; Supplementary Fig. 7). **F** Fluorescence recovery of the Cy3 donor presented in panel **D**, upon addition of a large excess of Polδ chemical trap (20 ng/μL of heparin) for the 2-nt gap (top; Sub#29) and the NP (bottom; Sub#28) substrates. The experimental datapoints were fitted to a dependence proportional to Eq. (1) ("Methods"). **G** Cartoon representation of the kinetic competition between FEN1 and Polδ for the NP from association rate perspective. Association rates ($k_{on}$) were determined from dissociation constants ($K_D$) and rates ($k_{off}$) based on the indicated equation. The $f_{i=1,2}$ coefficients represent the engagement probabilities of the NP by FEN1 and Polδ, respectively, based on their association rates and the indicated equation. Source data are provided as a Source Data file.

binding events would block the NP for ~5.4 s before Polδ is able to engage it and insert an additional nucleotide. Starting from a freshly cut NP, this estimation sets the shortest processing time of human NT to: 1.8 s for FEN1 dissociation (Fig. 5C), 5.4 s for the kinetic competition, ~0.1 s needed by Polδ to displace 1 nt[15], and ~0.3 s for the next FEN1 catalytic cycle (Supplementary Fig. 3D), yielding a total of >7.5 s. This value is in good agreement with the experimental NT processing time (~11 s/nt; Fig. 1F) and shows that, in humans, proceeding beyond the first SD-flap cleavage cycle is a slow and inefficient process mainly limited by re-engagement (association) of the NP by Polδ in the presence of FEN1 (see below).

The effect of FEN1 on NT processivity can also be addressed from a dissociation rate perspective. FEN1-mediated cleavage and dissociation from the NP require ~2.1 s (~0.3 s in Supplementary Fig. 3D plus ~1.8 s in Fig. 5C), while Polδ dissociation time from the NP is ~1.1 s (Fig. 5F). FEN1 might stabilize Polδ interactions within a FEN1•Polδ•PCNA•NP toolbelt complex by ~6-fold (Fig. 4B versus 5E), which would increase its dissociation time to <6.6 s (assuming its association

rate is unaffected). Following an exponential survival function for Polδ ($e^{-t*k_{off-Polδ}}$), this analysis indicates that up to ~27% of Polδ may dissociate during the very first catalytic cycle of FEN1, and this would happen for each 1 nt along the NT. Therefore, Polδ would completely dissociate after the removal of 4 nt by the NT, which is in agreement with the experimental 1–4-nt processivity (Fig. 1G).

## Additional mechanisms can accelerate human NT and MOF

The toolbelt model will give FEN1 immediate access to the NP and stop the next cycle of NT. This prompted us to investigate whether performing the NT reaction in a sequential manner that allows Polδ•PCNA to reengage the flap substrate in multiple cycles of SD and delay FEN1•Polδ•PCNA toolbelt formation will improve the NT rate. NT reactions were carried out for 30 s at fixed FEN1 and increasing Polδ concentrations (Fig. 6A). The NT reaction rapidly accelerated under Polδ:FEN1 ratios of up to ~4.5:1, whereafter only a minute increase was observed (Fig. 6B). At the breakpoint of the Polδ:FEN1 ratio, the NT removed a total of ~8.5 nt in 30 s. This rate approaches the rate of SD

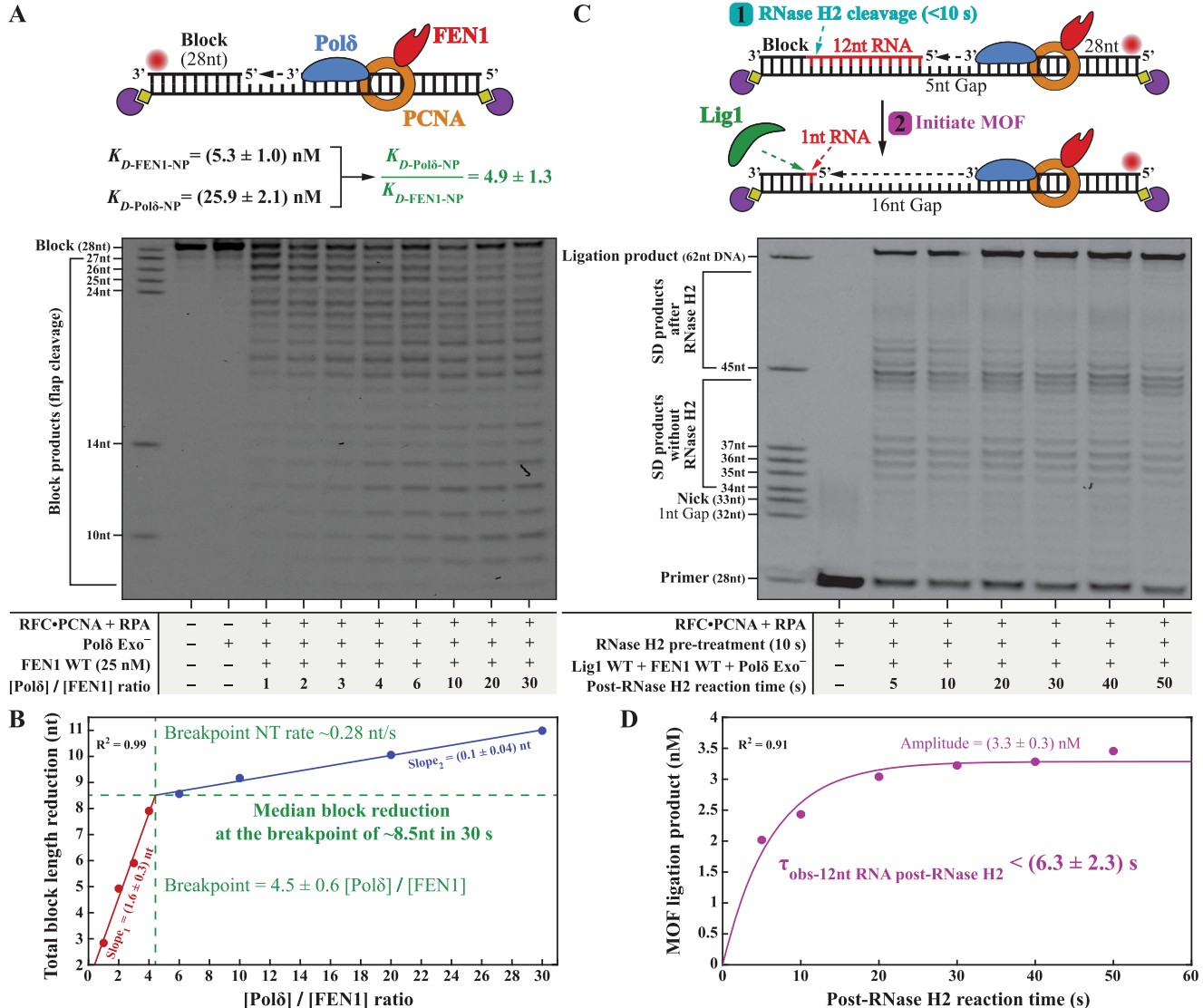

**Fig. 6 | Mechanisms that accelerate human NT and MOF. A** NT products monitored through the extension of the primer oligonucleotide at fixed FEN1 and varied Polδ concentrations: 25 nM FEN1, 37 °C, 30 s, DNA Sub#2 (Supplementary Fig. 7). **B** Quantification of the NT reaction products presented in panel **A**. Median product lengths beyond the first NP were determined for each Polδ:FEN1 ratio as described in the Methods section. The experimental datapoints were fitted to a continuous bilinear dependency with a variable breakpoint. The breakpoint rate was determined by dividing the total block length reduction at the breakpoint (~8.5 nt) by 30 s. The intercept of the model was determined from the fit as 1.3 ± 0.5 nt, probably

corresponding to 1-nt exonuclease product of FEN1 in the absence of Polδ (Supplementary Fig. 5). **C** MOF products monitored through the extension of the primer oligonucleotide following pre-treatment of the 12-nt RNA-containing block with RNase H2 (50 nM) at RT for 10 s: 250 nM Polδ, 250 nM FEN1, 250 nM Lig1, RT, DNA Sub#32 (Supplementary Fig. 7). **D** Quantification of the time dependence of the ligated MOF product yield after RNase H2 pre-treatment, from panel **C**. The experimental datapoints were fitted to a single-exponential product-formation burst equation [MOF Ligated Product = Amplitude $* (1 − e^{−t/\tau_{\text{obs–12nt RNA post–RNase H2}}})$]. Source data are provided as a Source Data file.

under similar conditions (Supplementary Fig. 1N), suggesting that, indeed, the flap is not engaged immediately by FEN1. Collectively, these results indicated that the minimum processing time for 12-nt RNA primer removal would be ~45 s under more sequential binding conditions that favor Polδ binding. This rate is ~4-fold faster than under strict toolbelt conditions, but it is still >15-fold slower than the yeast NT reaction[15].

While these results clearly demonstrate that the presence of FEN1 acts like a penalty for timely MOF completion, the aforementioned acceleration mechanism is rather unlikely in a cellular context due to the requirement of excess Polδ over FEN1. The quantitative proteome of a human cancer cell line[47] (nuclear volume[48] of ~1472 μm³) shows the contrary of this requirement; FEN1 (1.40 × 10⁵ copies/cell; ~158 nM) is considerably more abundant than Polδ p125 (1.96 × 10⁴ copies/cell; ~22 nM) or Lig1 (2.56 × 10⁴ copies/

cell; ~29 nM). These values also show that the typical protein concentrations of 25–250 nM employed in the current reconstitutions of the MOF, NT, and SD reactions are comparable to the nuclear concentration levels. Surveying the normalized expression levels of the MOF proteins across 253 tissue types (The Human Protein Atlas[49]) showed an average Lig1:Polδ p125 ratio of 1.0 ± 0.5 and an average FEN1:Polδ p125 ratio of 2.8 ± 1.5. In 47 cancer cell lines[50] the average FEN1:Polδ p125 ratio increased to 3.7 ± 2.2. Overall, this indicates average FEN1:Polδ:Lig1 nuclear ratios between 1:1:1 and 5:1:1. Extrapolating the results shown in Fig. 6B backward toward a Polδ:FEN1 ratio of 0.3:1 predicts a NT removal of ~2 nt in 30 s, only slightly faster than the exonuclease activity of FEN1 alone (Supplementary Fig. 5C, D). This suggests that under physiological FEN1:-Polδ ratios, NT beyond the first FEN1 catalytic cycle might be completely inhibited.

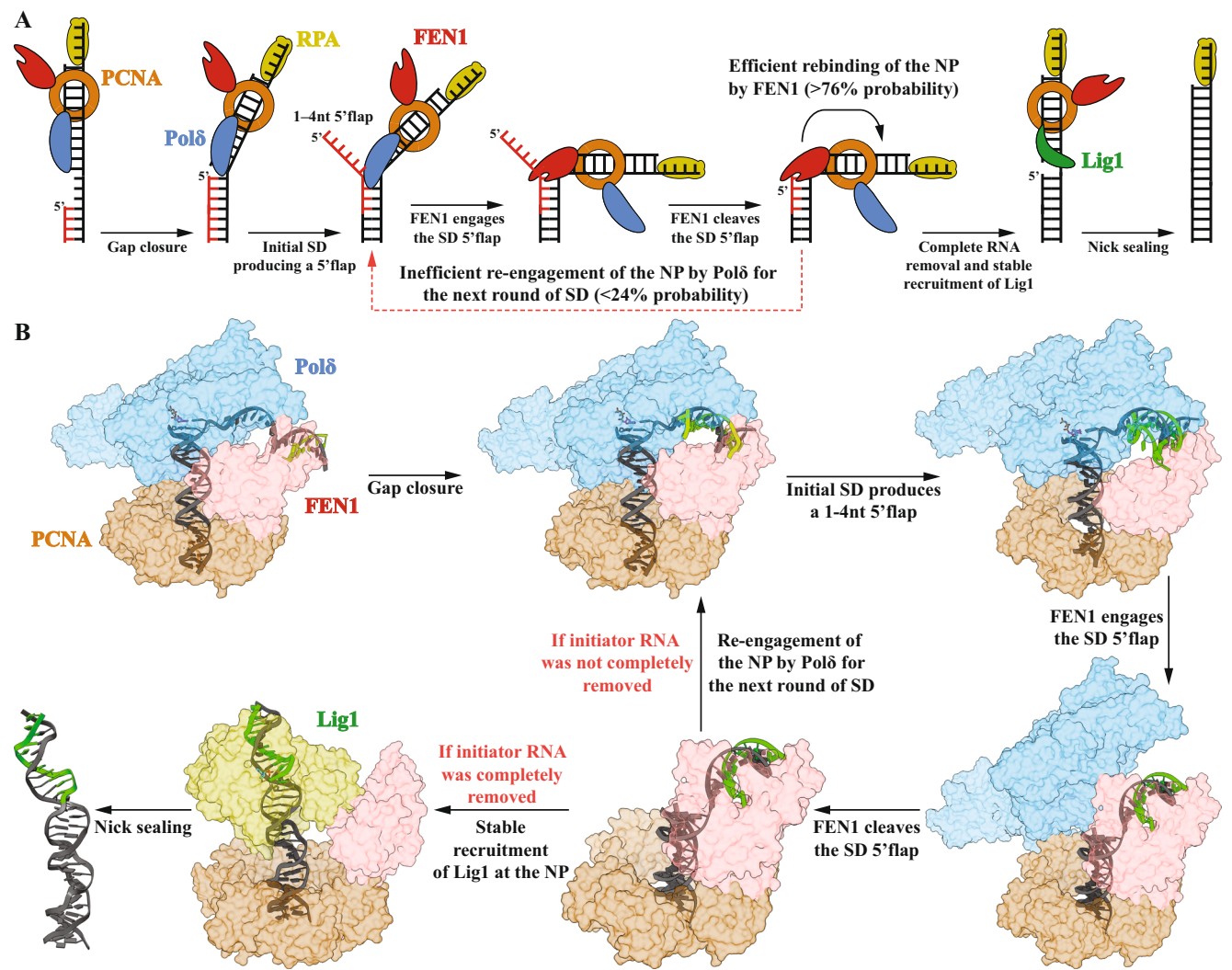

**Fig. 7 | Proposed model for human MOF. A** Cartoon and **B** structural model proposed for human MOF and its various intermediary structures and complexes. The structural model was generated based on the structural data presented in refs. 18, 20.

Last, we used human RNase H2 to investigate whether pre-removal of the RNA primer by RNase H ribonucleases[5,30,51–58] may accelerate MOF. These ribonucleases hydrolyze RNA moieties in RNA-DNA hybrids to leave behind exactly 1-nt RNA at the 5′ end of the OF. In a control experiment, we showed that the primer which exclusively contains DNA was unaffected by RNase H2, while the block containing 12-nt RNA was reduced to 1-nt RNA in <3.9 s (Supplementary Fig. 6A, B). Upon pre-treatment of the substrate with RNase H2, the reconstituted MOF reaction produced ligation products visible even at 5 s (Fig. 6C), and the overall reaction time was reduced to <6.3 s (Fig. 6D). Altogether, the times required for RNase H2 pre-cleavage (Supplementary Fig. 6B), for NT to remove the remaining 1-nt RNA (Fig. 1F), and ligation (Supplementary Fig. 1B) lead to a total MOF time of ~15 s, ~10-fold faster than without RNase H2 involvement (>150 s in Fig. 1C).

## Discussion

The current study provides a detailed mechanistic understanding of the kinetics and substrate hand-off mechanisms during human MOF as well as of how PCNA orchestrates the activities of Polδ, FEN1, and Lig1 through two dynamic toolbelts. The formation of these toolbelts is not limited by the binding to PCNA but rather is regulated by competition for DNA. This competition favors Polδ binding to the P/T for OF synthesis, FEN1 to the flap substrate for cleavage, and Lig1 to the NP for ligation if the RNA was completely removed (Figs. 2E and 7A). The

study also highlighted how the lower stability of human Polδ•PCNA on DNA relative to yeast[23] compromised substrate hand-off from FEN1 to Polδ to result in a NT reaction that is ~55-fold slower and less processive[15]. Therefore, in yeast, the toolbelt mechanism enhances NT reaction rate and processivity as compared to SD, while in humans it acts as a penalty. In fact, deviation from a strict toolbelt NT reaction that allows Polδ•PCNA to perform multiple cycles of SD before PCNA recruits FEN1 and forms the FEN1•Polδ•PCNA toolbelt increased the rate by ~4-fold. However, the fastest pathway for MOF in humans requires pre-removal of the RNA primer such as by RNase H2, which leaves behind exactly 1-nt RNA that can be further removed in a single NT cycle and then ligated to complete MOF in ~15 s.

Combined with recent structural data, these findings provide a model for human MOF (Fig. 7B). The toolbelt structure of human FEN1•Polδ•PCNA bound to P/T shows that the template strand is bent by 90 degrees and projects toward FEN1[18]. This suggests that when Polδ finishes a nascent OF and starts to displace the primer of the previous OF, FEN1 will likely face and bind downstream dsDNA, which may facilitate transfer of the flap substrate from Polδ to FEN1. In fact, our single-molecule experiments indicate that Polδ hands off a bent flap substrate to FEN1. The toolbelt structure also suggests that either Polδ or FEN1 can bind the PCNA-encircled upstream dsDNA[18]. Therefore, upon 5′flap cleavage, the stable complex of FEN1•PCNA with the NP will sequester the upstream dsDNA and promote dissociation of the

weakly PCNA-bound Polδ. In the case of yeast, the 50% more contacts of Polδ with PCNA[19] may structurally favor the handover of the FEN1-bound NP to Polδ, by forcing Polδ into a conformation that brings it closer to the PCNA-encircled upstream dsDNA. Upon Polδ dissociation, Lig1 would bind the free site on PCNA to form a second toolbelt with the FEN1-bound NP[20] and force the release of the NP. If RNA were not completely removed, the high fidelity of Lig1 against 5′ RNA would lead to its dissociation, re-association of Polδ, and finally the restart of the next NT reaction cycle.

Since each Polδ molecule must be reused ~1220–3060 times per S-phase[5,47], it is highly unlikely that unaided human NT will proceed beyond the first processive cycle of 1–4 nt even at long time scales. While proceeding beyond this first cycle is inefficient and requires complete re-association of Polδ, recent in vivo data show that human Polδ localizes for extended periods of time, up to tens of seconds, at certain replication foci that were associated with MOF activity[59]. Therefore, it is possible that at least in a fraction of OFs, RNA removal may proceed slowly and inefficiently, as described in the current study. However, it remains possible that additional pathways and/or interactions might aid in accelerating MOF. For example, additional interactions with the replisome may stabilize Polδ in the NT complex and force its DNA re-engagement. Additional pathways might also involve the pre-removal of initiator RNA before the arrival of the NT complex. Both human RNase H1[60] and H2[61] perform very fast cleavage in <4 s that can occur simultaneously with current OF synthesis that requires >2 s[23]. We showed that in this scenario, RNase H2 cleavage followed by the removal of the remaining 1-nt RNA by a single cycle of NT reaction and ligation can complete MOF in ~15 s. Interestingly, RNase Hs are not required for viability or even efficient cellular proliferation in yeast[62], while their mutations or deletions result in several abnormalities or even lethality in higher vertebrates[63–66]. In addition, in vitro reconstitutions of mouse cell[67] and simian virus 40[68] DNA replication proposed that RNase H activity is required for efficient MOF. Remarkably, RNase H acts as a processive exoribonuclease on unmatured OF-like substrates[58], which would predominantly degrade the initiator RNA into mono-ribonucleotide products, similar to NT[15] and probably to in vivo MOF products. Other pathways may involve recruiting proteins that work with Polα to pre-remove the initiator RNA immediately after primer synthesis[8]. For example, Polα interacts and stimulates FEN1 5′-to-3′ exonuclease activity[69], and similar interactions of Polα with RNase Hs have been reported in various organisms[70]. Finally, our study does not address the effect of post-translational modifications on the NT reaction. Phosphorylation of human FEN1 is known to decrease its interaction with PCNA[71], which may promote dissociation of FEN1 from the NP and its more efficient re-engagement by Polδ. It may also provide a mechanism that promotes our reported sequential SD and flap cleavage mechanism to enhance the rate of MOF. In addition, post-translational modifications might increase Polδ stability to the point where it can actively displace FEN1.

While the RNase H pathway can completely compensate the inefficiency of the human NT reaction for initiator RNA removal, a second limitation would appear in removing the Polα-synthesized 10–20-nt DNA region (termed α-segment[72]) of the primer. As a proofreading-deficient DNA polymerase, Polα is expected to incorporate errors into α-segments. In yeast, in the absence of Lig1 intervention, the processive NT reaction could itself remove and resynthesize the entire α-segment[15]. However, in human even in the presence of RNase Hs, the NT reaction would processively remove only the first 0–3 nt of the α-segment. Indeed, it was shown that the DNA synthesized by Polα is considerably retained in vivo into the mature genome[73]. Several possibilities can be considered to reconcile the low DNA-region invasion by NT with low lagging strand mutation rates. First, additional interactions between the NT complex and the replisome may stabilize Polδ for promoting NT further into α-segments. Second, human Polα may intrinsically exhibit

reduced error rates as compared to its counterparts from lower eukaryotes. Last, it is possible that the correction of Polα-generated errors compulsorily involves DNA repair pathways[74]. In fact, near the 3′ end of the α-segments, it has been suggested that the proofreading activity of the newly assembled Polδ holoenzyme can sense and correct Polα-generated errors[75]. On the other hand, the 5′ end of α-segments represents a strong signal for mismatch repair (MMR), a pathway that is particularly active on the lagging strand[76]. A specialized type of MMR that uses FEN1 as exonuclease, without requiring Exonuclease-1 (Exo1), called α-segment error editing (AEE)[72], has been shown to correct mismatches arising within <12 nt from the 5′ end of α-segments. Polα interacts directly with MutSα[77], the complex responsible for mismatch and 1–2-nt indel recognition in MMR[78], which open the possibility that error sites in the α-segments might be prebound by MutSα before the arrival of the NT complex. In AEE, FEN1 and MutSα form a unique complex that dramatically stimulates FEN1 exonuclease activity post-RNA-removal on mismatch-containing α-segments[72]. Therefore, on such segments, FEN1 from the incoming NT complex might engage promptly with the prebound MutSα to initiate AEE for immediate repair. Taken together, the extrinsic correction via Polδ proofreading activity and the intrinsic correction via AEE can efficiently remove errors from entire α-segments[79], without the need for NT invasion into the DNA. Finally, any remaining errors may be corrected by additional DNA repair pathways (e.g., traditional MMR), probably after OF ligation.

Apart from their role in DNA replication, Polδ's gap-filling and SD activities are also fundamental for a variety of DNA repair and recombination pathways[80,81]. The intrinsically lower stability of human Polδ and the inhibition of its SD activity by FEN1, point toward fundamental biological differences between lower and higher eukaryotes. In translesion DNA synthesis, the lower stability of human Polδ can enhance its switching to translesion DNA polymerases without the stringent requirement for PCNA ubiquitination. In Exo1-dependent MMR, short-patch base excision repair (BER), and nucleotide excision repair, the suppression of Polδ's SD activity by FEN1 may lead to a timely completion of the NP ligation without excessive SD progression into the healthy strand. Oppositely, Exo1-independent MMR, long-patch BER, and homologous recombination-mediated double-strand break repair, require considerable SD activity by Polδ, and therefore may require further mechanisms to prevent efficient FEN1 recruitment. Collectively, our results demonstrate that human cells evolved to lower the stability of Polδ on DNA in order to control its SD activity. These findings call for investigating the implication of this control mechanism during DNA repair and recombination in higher eukaryotes.

Collectively, our results propose that, in humans, pre-removal of the RNA primer is the most efficient MOF pathway, requiring ~15 s. This is followed by sequential SD and flap cleavage NT reactions that can achieve MOF in ~45 s. The unaided toolbelt pathway would be the slowest, requiring ~155 s. However, both the sequential and toolbelt pathways are un-processive, requiring multiple dynamic association and dissociation cycles especially for Polδ. These mechanisms are mediated by at least two distinct dynamic toolbelt complexes. Nonetheless, it is possible that these toolbelts form only transiently during the initial stages of substrate handover from Polδ to FEN1 and then directly to Lig1 if RNA pre-removal was successful. Last, our findings call for extreme care when extrapolating results from lower to higher organisms, even despite a common eukaryotic lineage.

## Methods
### Protein expression, purification, and labeling
Wild-type proteins and their mutants were expressed and purified as described previously. Human FEN1 WT (amino acids: 2−380), FEN1 D181A, FEN1 S293C, and FEN1 ΔC (amino acids: 2−336) were expressed in *E. coli* and purified using the SUMO system as described in[13,36]. N-terminally His-tagged PCNA WT and PCNA N107C were expressed

in *E. coli* and purified as described in[82,83]. Tag-free Lig1 WT was expressed in *E. coli* and purified as described in[84]. Lig1 ΔN (amino acids: 233−919) was expressed as a SUMO fusion protein in *E. coli* and purified as described in[13,36,85] for SUMO fusions. Polδ WT, Polδ Exo⁻ (D515V in p125), Polδ Exo⁻ Pol⁻ (D515V, D602A, D757A, and E795A in p125), and Polδ p12⁻ were expressed in Sf9 insect cells and purified as described for Polδ WT in ref. 18. His-tagged RPA and ΔN-RFC (amino acids: 554−1148 in RFC₁) were expressed in *E. coli* and purified as described in ref. 18. DNA2 was purified as previously described in ref. 86, with the following modification: the C-terminal Flag-tag was replaced with a C-terminal Strep-tag for the introduction of a Strep-affinity chromatographic step. RNase H2 was purified as previously described in ref. 61. Unless otherwise specified, Polδ refers to Polδ Exo⁻. For simplicity, ΔN-RFC is referred to as RFC. For the rest of the proteins, we refer to their WT version, unless otherwise specified.

FEN1 S293C was labeled with Cy3 maleimide (Lumiprobe) to a final stoichiometry of 1:1 Cy3 to FEN1. PCNA N107C was labeled with Cy5 maleimide (Lumiprobe) to a final stoichiometry of 2.7:1 Cy5 to PCNA trimer. For both labeled proteins, the chemical labeling reactions and free-dye removal steps were carried out identically, as described in ref. 87.

## DNA and RNA oligonucleotides and substrates

All oligonucleotides were purchased from Integrated DNA Technologies (IDT) and were HPLC-purified by the manufacturer. All oligonucleotides used for generating RNA-containing substrates were purified by the manufacturer via RNase-free HPLC purification.

Substrates were annealed by mixing their component oligonucleotides at specific ratios in a buffer containing 50 mM TRIS pH 8.0, 10 mM EDTA, and 100 mM NaCl (TE 100) and then heated at 95 °C for 5 min, followed by slow cooling to room temperature in a thermocycler. The oligonucleotide mixing ratios used for substrate annealing before gel purification are indicated in Supplementary Fig. 7 for each substrate. Throughout the manuscript, substrates are numbered according to the codes presented in Supplementary Fig. 7. All substrates were purified on 10% non-denaturing TBE-PAGE gels and recovered from gel by the crush and soak method in TE 100 for 30 min at 16 °C and vigorous shaking. Substrates used in the experiments presented in the current manuscript exhibited >80% purity. For the bulk experiments that used blocked biotinylated substrates, a 2.5× molar excess of tetrameric NeutrAvidin (GE Healthcare) was added to the substrates and incubated for 5 min prior to the experiments.

## SD, NT, and MOF reconstitution reactions and analysis on short substrates

All SD, NT, and MOF reactions were performed in a reaction buffer containing 50 mM HEPES-KOH pH 7.5, 5% (v/v) glycerol, 1 mM dithiothreitol (DTT), 0.1 mg/mL bovine serum albumin (BSA), 100 mM KCl, 10 mM MgCl₂, and 1 mM ATP. DNA substrates (10 nM) blocked with 25 nM NeutrAvidin were used in all SD, NT, and MOF reactions. SD and NT reactions were carried out at either room temperature (RT) or 37 °C for different amounts of time, as indicated in each case. For SD reactions, NeutrAvidin-blocked DNA, 50 nM RPA, 40 nM PCNA, 20 nM RFC, and 25−250 nM Polδ Exo⁻ were pre-incubated in reaction buffer for 1.5 min. Reactions were initiated via the addition of 500 μM dNTPs. For the unsaturated NT reactions, NeutrAvidin-blocked DNA, 50 nM RPA, 40 nM PCNA, 20 nM RFC, and 25 nM Polδ Exo⁻ were pre-incubated in reaction buffer for 1.5 min. Thereafter, the reactions were initiated through the addition of 500 μM dNTPs and 25 nM FEN1. In reactions containing Polδ WT or Polδ p12⁻, 250 μM dTTP was also added during the pre-incubation step to avoid Polδ exonuclease activity, and then the reactions were initiated through the addition of 500 μM dNTPs. For the fully saturated NT reactions, NeutrAvidin-blocked DNA, 50 nM RPA, 60 nM PCNA, 40 nM RFC, 250 nM Polδ, and 250 nM FEN1 were used.

MOF reactions were carried out as descried for the fully saturated NT reactions, but were initiated through the addition of 500 μM dNTPs, 250 nM FEN1, and 250 nM Lig1.

For MOF reactions containing RNase H2, the NeutrAvidin-blocked DNA substrate loaded with the Polδ holoenzyme was pre-treated with RNase H2 (50 nM) for 10 s before initiating the reaction with 500 μM dNTPs, 250 nM FEN1, and 250 nM Lig1.

All reactions were quenched by the addition of 40 mM EDTA, treated with proteinase K at 50 °C for 15 min, and stopped by adding an equal volume of stop buffer (50 mM EDTA, 95% formamide). DNA in the quenched reactions was denatured by heating at 95 °C for 5 min and then immediately placed on ice. DNA reaction products were separated on 20% denaturing Urea-PAGE gels and visualized using Typhoon Trio (GE Healthcare).

For all reconstituted reactions, the median length of the reaction products at a particular timepoint was estimated using a modified version of the median analysis presented in refs. 15, 88. At each experimental timepoint, product intensities were integrated between the 1-nt SD (or NT) product and the top or bottom of the corresponding lane. The position ($R_f$ value) corresponding to 50% integrated synthesis was then obtained. A calibration curve ($R_f$ value vs. oligonucleotide length) was initially generated for each gel. The position of median synthesis was back-translated into nucleotides by using the calibration curves. Pointwise processivity factors (insertion probabilities) were calculated per each lane of interest as described in ref. 23, based on Eq. (S21) (Supplementary Methods). Insertion probabilities were converted to survival probabilities using Eq. (S22) (Supplementary Methods) and fitted to exponential decay survival functions as per Eq. (S26) (Supplementary Methods).

## SD and NT reconstitution reactions on long forked-duplex substrates

The 2.8 kbp primed forked linear DNA was prepared as previously described in ref. 22. Reactions contained 30 nM Polδ WT, 30 nM RFC, 200 nM PCNA, 600 nM RPA, and 8 nM linear forked template in a reaction buffer consisting of 50 mM HEPES pH 7.5, 5% Glycerol, 0.1 mg/mL BSA, 1 mM DTT, 2 mM TCEP, 10 mM MgCl₂, 100 mM KCl, 2 mM ATP, and 150 μM of each dNTP. The assay was performed by pre-assembling Polδ WT, ΔN-RFC, and PCNA with the linear forked DNA in the presence of dATP and dCTP for 2 min at 37 °C. Reactions were started by the addition of RPA, dGTP, dTTP, and the indicated amount of FEN1 and/or DNA2 for the indicated time at 37 °C. Reactions were quenched upon the addition of 40 mM EDTA and analyzed in 0.6% alkaline agarose gel (30 mM NaOH and 2 mM EDTA) at 15 V for 17 h. The gel was backed with DE81 paper, compressed, and imaged in a Sapphire Biomolecular Imager (Azure Biosystems).

## Bulk-fluorescence measurements

Fluorescence emission spectra were measured using a Fluoromax-4 (HORIBA Jobin Yvon) spectrofluorometer equipped with a temperature control unit and a magnetic-stirring cuvette holder. The spectrofluorometer was controlled using the dedicated FluorEssence software (Horiba). All measurements were performed in a reaction buffer containing 50 mM HEPES-KOH pH 7.5, 5% (v/v) glycerol, 1 mM DTT, 0.1 mg/mL BSA, 100 mM KCl, 10 mM MgCl₂, and 1 mM ATP. Samples were excited at 530 nm, and emission was collected from 550 to 750 nm, with an increment of 1 nm and an integration time of 0.2 s. Both emission and excitation slits were set to 5 nm, and a 550 nm filter was placed on the emission side to prevent excitation light leakage into the emission pathway. The temperature was maintained at 22 °C. Emission and excitation polarizers were set to 0° and 54.7°, respectively (VM configuration), to eliminate polarization anisotropy effects.

FRET spectra were blank-corrected and normalized as described in ref. 89. For calculating apparent FRET efficiencies, donor emission intensity (D) was integrated from 560 to 580 nm, and acceptor

emission intensity (A) was integrated from 660 to 680 nm. Apparent FRET efficiencies were calculated as E = A/(D + A). For PIFQ, and FeSQ experiments, the spectra were blank-corrected as described in[36], but not normalized. For FRET experiments with FEN1-Cy3 and DNA-Alexa647, the spectra were neither blank-corrected nor normalized but fitted with a linear combination of Cy3 and Alexa647 spectra.

Typical FRET, PIFQ, and FeSQ experimental concentrations were 10 nM DNA, 100 nM RPA, 100 nM NeutrAvidin, 1 mM ATP, 100 nM PCNA, 100 nM RFC, 250 nM Polδ Exo⁻, 500 nM Lig1, and 15 nM FEN1 WT (or D181A), unless otherwise indicated or unless a titration was performed. For titration experiments, all binding isotherms were fitted to equations proportional to parabolic dependencies[90] as:

$$ES(E_0) = 1/2 \left[ E_0 + S_0 + K_D - \sqrt{(E_0 + S_0 + K_D)^2 - 4E_0 S_0} \right] \quad (1)$$

where $S_O$ and $E_O$ denote the total substrate and enzyme concentration, respectively, $ES$ denotes the enzyme-bound substrate, and $K_D$ denotes the dissociation constant. This equation was scaled in amplitude by specific parameters depending on the experimental signal followed during titration. For each point of the experimental isotherms, at least 1 min of incubation was allowed for binding equilibrium to be reached.

The fluorescence recovery experiments used to measure FEN1 and Polδ dissociation rates were performed under constant stirring with a small magnet inserted in the cuvette. Samples were excited at 530 nm, and donor (Cy3) emission at 565 nm was monitored over time, with a temporal resolution of 50 or 100 ms (integration time). Both excitation and emission slits were opened to maximum width, and a 550 nm filter was placed on the emission side. Excitation and emission polarizers were set to VM configuration. After ~10 s of signal stability, the trap competitor was suddenly injected into the cuvette, and signal acquisition continued for an additional ~30 s. The concentrations used for fluorescence recovery experiments were 50 nM DNA, 100 nM RPA, 100 nM NeutrAvidin, 1 mM ATP, 100 nM PCNA, 100 nM RFC. FEN1 WT (or D181A) was used at a final concentration of 50 nM, while its trap (DNA double flap substrate containing an unpaired 3′ nucleotide, and a completely phosphothiolated 5′flap oligonucleotide to enhance binding while eliminating catalysis) was injected at a final concentration of 5 μM. Lig1 competitor of FEN1 to the NP was injected at a final concentration of 2 μM. Polδ Exo⁻ was used at a final concentration of 50 nM, while its trap (heparin polysaccharide) was injected at a final concentration of 20 ng/μL. Signals were normalized to an average intensity of 1 arbitrary unit in the baseline region before trap addition. The fluorescence recovery burst phase was fitted to a single-exponential burst equation as:

$$F(t) = A - B * e^{-t * k_{off}} \quad (2)$$

where A and B are constants related to the fluorescence intensity (F) before and at the end of the fluorescence recovery transition, while $k_{off}$ is the protein dissociation rate from the labeled DNA substrate.

## Single-molecule imaging and analysis

Glass coverslips were functionalized by 1:100 molar ratio of biotinylated PEG to mPEG[91]. The coverslips were used for building a flow cell with one inlet and one outlet tubing, as previously described[13,14,36,91]. Before each experiment, 0.2 mg/mL NeutrAvidin were injected into the flow cells and incubated for 10 min. Excess NeutrAvidin was removed by extensive washing with reaction buffer. The reaction buffer contained 50 mM HEPES-KOH pH 7.5, 5% (v/v) glycerol, 1 mM DTT, 0.1 mg/mL BSA, 100 mM KCl, and 10 mM MgCl₂. The reaction buffer was adjusted to a final pH of 7.5 after the addition of all components by KOH. Biotinylated DNA substrates were diluted from the −20 °C aliquoted stocks to a final concentration of ~200 pM in reaction buffer. The diluted substrate solutions were filtered through syringe filters with 0.2-μm pore size and

then injected into the flow cell until an optimal coverage of ~200 molecules/field of view was achieved. The unbound excess substrate was removed by extensive washing with reaction buffer.

To aid fluorophore photostability for imaging, reaction buffer was supplemented with an oxygen scavenger system (OSS) and a reduction–oxidation triplet-state inhibitor system (ROXS). The OSS[92] was composed of 6 mM proto-catechuic acid (PCA) and 60 nM proto-catechuate-3,4-dioxygenase (PCD). PCD enzyme was custom-purified as described in ref. 36. The OSS system enzymatically eliminates oxygen from the imaging buffer to prevent fluorophore photobleaching. The ROXS system contained 2 mM Trolox/Trolox-Quinone[93] (~80% TX and ~20% TQ mixture obtained through slow aging at 4 °C). ROXS is intended to minimize fluorophore photoblinking and therefore, indirectly, also photobleaching. Both, the OSS and ROXS component concentrations represent the final concentrations used in the imaging buffer.

TIRF-based FRET imaging was performed using a custom-built set-up described previously[91,94]. Movies were acquired using xCellence rt (Olympus) under continuous excitation (CW) with a green laser, at a temporal resolution of 50, 100, or 160 ms, as indicated for each experiment. A transformation matrix file was generated by imagining fluorescent beads to aid particle linking between the green and red emission channels, as previously described[91]. All movies were analyzed using the *TwoTone* software integrated into MATLAB (MathWorks) as previously described[13,14,36,91,95].

FEN1 cleavage assays that did not include PCNA were performed as described in refs. 13, 14, 36. Those that included PCNA loaded by RFC and trapped on the substrate, were performed as described hereafter. A PCNA loading solution was prepared by adding 30 nM PCNA, 15 nM RFC, 10 nM RPA, and 1 mM ATP into the reaction buffer. This solution was incubated for 1 min at 37 °C before injecting it into the DNA-containing flow cell. Following injection, the solution was further incubated in the flow cell for another 1 min, and then the excess unbound proteins were removed via extensive washing with reaction buffer containing 10 nM RPA. RPA was then maintained in all the solutions injected into the flow cells. For all single-molecule experiments, FEN1 and Polδ Exo⁻ were used at a final concentration of 250 nM.

Particles with either the donor or the acceptor missing were discarded by default in the *TwoTone* software, as these particles failed to participate in the particle linking step. In addition, we discarded the traces that showed aberrant FRET values in the substrate alone phases, traces with the extreme level of noise, as well as traces with atypical total emission intensity that may indicate the presence of multiple donors or acceptors.

The dwell times corresponding to a particular FRET state of interest were manually determined by counting the number of frames associated with that FRET state and by considering the experimental temporal resolution.

## FEN1 cleavage, RNase H2 digestion, and Lig1 ligation kinetics assays

All reactions were performed in a reaction buffer containing 50 mM HEPES-KOH pH 7.5, 5% (v/v) glycerol, 1 mM DTT, 0.1 mg/mL BSA, 100 mM KCl, 10 mM MgCl₂, and 1 mM ATP.

FEN1 multiple turnover cleavage assays on flap substrates included 500 nM DNA and 1 nM FEN1 WT (or FEN1 ΔC). Reactions were initiated by FEN1 addition and incubated at 37 °C for the indicated amount of time.

FEN1 exonuclease cleavage assays on gap and nick substrates included 10 nM NeutrAvidin-blocked DNA, 50 nM RPA (only for the nick substrate), 40 nM PCNA, 20 nM RFC, and 250 nM FEN1 WT. Reactions were initiated by FEN1 addition and incubated at 37 °C for the indicated amount of time.

RNase H2 cleavage assays on gap substrate containing 12-nt RNA included 10 nM NeutrAvidin-blocked DNA and 250 nM RNase H2.

Reactions were initiated by RNase H2 addition and incubated at 37 °C for the indicated amount of time.

Lig1 multiple turnover assays included 250 nM DNA and 1 nM Lig1 WT (or Lig1 ΔN) in the reaction buffer. Reactions were initiated by Lig1 addition and incubated at 37 °C for the indicated amount of time.

All the reactions were quenched and treated with proteinase K, and the DNA products were denatured and visualized as described above for the SD, NT, and MOF reactions. Products were quantified as a percentage of product(s) intensity divided by total lane intensity.

## Protein–protein electrophoresis mobility shift assay

EMSAs were conducted in a reaction buffer containing 1 nM PCNA-Cy5 with increasing concentrations of Polδ Exo⁻ or Lig1. Reaction mixtures were incubated at RT for 20 min, then 5% (v/v) Ficoll was added to the reactions, and the entire reaction volume was loaded onto 6% non-denaturing TBE-PAGE gels. The gels were run for 1 h at RT at 70 V in TBE buffer. Gels were visualized using Typhoon Trio (GE Healthcare). The percentage of free PCNA was estimated by dividing the intensity of the band corresponding to free PCNA in each lane by the intensity of the band corresponding to PCNA in the lane that contained no interaction partner. The percentage of bound PCNA was estimated by subtracting the percentage of free PCNA from the total. All binding isotherms were fitted to parabolic dependencies as described above (Eq. (1)).

## Data analysis and plotting software

All the data presented in the current study were analyzed and plotted using the OriginPro and MATLAB software.

## Statistics and reproducibility

All the experiments presented in Fig. 1B, D, E, G, H; 6A, C and Supplementary Figs. 1A, E, F, K–M, P, Q, T, U; 2E, F; 5A, C and 6A were repeated independently at least three times with reproducibility of >80% in terms of median products lengths and/or product yields.

## Reporting summary

Further information on research design is available in the Nature Portfolio Reporting Summary linked to this article.

# Data availability

The data supporting the findings of this study are available within the article and its supplementary information files. Any additional information supporting the findings of this manuscript is available from the corresponding authors upon reasonable request. The cellular expression levels of the human MOF proteins were estimated from the datasets presented in The Human Protein Atlas (http://www.proteinatlas.org) for Polδ p125 with identifier ENSG00000062822, for FEN1 with identifier ENSG00000168496, and for Lig1 with identifier ENSG00000105486. Source data are provided with this paper.

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

## Acknowledgements

This work was supported by the King Abdullah University of Science and Technology under Competitive Research Award Grant CRG8 URF/1/4036-01-01 to S.M.H. and A.D.B. We thank Yujing Ouyang for the preparation of the functionalized coverslips. We also thank the members of Samir M. Hamdan's lab for the helpful discussions. We thank Prof. Marc S. Wold for the generous gift of human RPA expression plasmid. We thank Prof. Petr Cejka for the generous gift of human DNA2 expression plasmid. We thank Prof. Andrew Jackson and Dr. Martin Reijns for the generous gift of human RNase H2 expression plasmid (pGEX6P1-hsRNASEH2BCA, Addgene plasmid #108692).

## Author contributions

S.M.H., V.S.R., and M.T. initiated the project. V.S.R., M.T., and S.M.H. designed, planned, and analyzed the experiments. V.S.R. and M.T. performed the experiments. A.A. performed the experiment on the long DNA substrate. V.S.R. performed the mathematical modeling and data fitting. M.T. established the expression and purification for some of the proteins used in the current study in our laboratory. L.J. and V.S.R. performed preliminary experiments on the interaction of FEN1 with PCNA. V.S.R., M.T., A.D.B., and S.M.H. discussed and interpreted the results, and wrote the manuscript. V.S.R., M.T., A.D.B., and S.M.H. revised and edited the manuscript. All authors discussed the results and commented on the manuscript.

## Competing interests

The authors declare no competing interests.
