## [Peer Review File · Nature Communications]

Unaided Human maturation of Okazaki fragments is dynamic, non-processive and strikingly slowREVIEWER COMMENTS

Reviewer #1 (Remarks to the Author):

In this manuscript, Raducanu and co-workers use a variety of biochemical assays to investigate the dynamics of Okazaki-fragment maturation with purified human proteins. The authors find that this maturation is really quite slow *in vitro*, and that the processivity of DNA polymerase delta is substantially reduced by competition with FEN1 for nicked or flap substrates. The authors demonstrate that, while Pol delta, FEN1 and DNA ligase can apparently all associate with PCNA on a replication template simultaneously, FEN1 outcompetes Pol delta even after cleaving a flap to regenerate a nick: in turn, FEN1 can be actively removed from a nicked DNA substrate by DNA ligase 1. For lagging-strand substrates with several ribonucleotides at their 5' ends, nick translation must proceed via repeated cycles of alternating Pol delta/FEN1 activity, with this nick translation rendered slow by the preferential binding of FEN1 (as opposed to Pol delta) to nicked substrates. Pre-removal of the RNA primer by RNase H2 leads to faster Okazaki fragment ligation after only a few rounds of nick translation.

Overall, I think this is a very strong piece of work that provides important insights into a fundamental process, and will be of great interest to researchers in the DNA replication field. The data are of high quality, and are clearly presented and described throughout the manuscript: the authors have also been careful not to overstate their findings, and do a nice job discussing potential caveats in the discussion. I strongly recommend publication of this work.

I think that the manuscript is acceptable for publication in its current form, but have a couple of minor suggestions that the authors could address.

The result shown in Figure S1P, namely that Dna2 and FEN1 together inhibit strand-displacement synthesis to a greater extent than FEN1 alone, is very interesting and I would advocate moving it from the supplement to the main figures.

The data in figure 4 appear to assume that all proteins under investigation remain associated: if this is the case, it would be helpful to clarify and to provide a rationale for this assumption. If such an assumption cannot be made, the authors should state this caveat.

The title of the manuscript is a bit overblown: I think something slightly more descriptive and reflective of the results would be appropriate. As a side note, given that this study raises real questions about how human cells accomplish Okazaki fragment processing, one might argue that the work in fact does the opposite of demystify the process.

The data presented in the manuscript suggest that, in the absence of substantially different reaction kinetics *in vivo*, the lagging strand would invariably contain a large amount of DNA synthesized by a proofreading-deficient DNA polymerase (Pol alpha). Although the authors do a nice job highlighting limitations and potential caveats inherent in their biochemical work, the idea that pol alpha contributes significantly to the mature lagging-strand would be hard to reconcile with low mutation rates. I think the authors should address this issue in the discussion.

Reviewer #2 (Remarks to the Author):

This is an exciting paper about Maturation of Okazaki Fragments (MOF) that addresses some important questions in the field about the enzymatic order-of-operations with precise single-molecule fluorescence experiments. The subject and general findings could ostensibly be suitable for publication in Nature Communications after revision. Generally, the bulk biochemical section is difficult to follow,

and needs special attention to make it readable. However, the bulk and single-molecule fluorescence experiments are comprehensively explained, and the interesting conclusions regarding substrate hand-off during MOF are supported. Generally, sufficient information is provided to allow reproduction. The authors converge on a holistic mechanistic model of MOF using evidence from relative affinities, competition assays, and single-molecule kinetics, and structural information. The fact that RNAseH sped of MOF kinetics by such a degree is also of great interest. The discussion is thorough, the model is nicely presented, and overall, the data in theory would be well received in the field once the following issues are addressed:

Major Critiques

1. The first section is particularly difficult to follow. In general, please explain the experimental design and the calculations involved explicitly, with careful attention to maximizing reader comprehension.
2. Line 86, an "apparent MOF" is not defined as a variable in the results section. Looking at the figure, a parameter called $T_{\text{obs-12nt RNA}}$ seems to be the same as MOF, but is also not defined (a similar variable needs to be defined in 6D). To try and explain how the data was fit, the figure legend of 1C calls out Eq. 2 of Materials and Methods, but this equation is not relevant. I would have guessed you wanted to point to Eq S15, but neither of these variables are named in your kinetic model either. The "Derivation of a fitting model for polymerization kinetics data" section in Supplementary Methods needs an introductory paragraph for lay audiences. Explain more simply at first why the model needed to be developed. If there are differences between your model and other extant models in the literature, please explain those here, along with your rationale for following your methods.
3. Line 87: "Since the NT reaction did not proceed considerably beyond the 12nt RNA (Figure 1B), the processing time for RNA removal was estimated to be ~ 13 s/nt." It's difficult to understand from this statement where the 13 s/nt estimate for RNA removal was calculated from.
4. In Figure 1H, the mechanism of heparin trapping is unclear. How is heparin specific for DNA-unbound Pol d? Please either perform the experiment using an unlabeled T/P pair as a trap instead or explain why heparin is superior.
5. It is unclear why the bulk DNA substrates are blocked with neutravidin and is only mentioned in passing. Wouldn't this slow down the kinetics of OF maturation? Or was it for the purposes of comparison to the TIRF experiments? Is it to keep PCNA bound? Please clearly explain rationale.
6. Fig 3C/D. It is unclear in the text why PCNA binding might (or might not) matter. What is the context of this experiment? Also, the mean calculation and/or curve fit are not explained in the legend. It almost appears as though a distinct population arose from PCNA binding in 3C. Could the authors please discuss and demonstrate empirically why this should not be fit to a bimodal distribution?

Minor Critiques

1. The title is vague enough to read like the title of a review. Please consider changing it to directly reflect your findings.
2. Line 21 "They also suggest.."
3. Line 29, Refs 8 and 9 shouldn't be separated as they both address RNA/DNA synthesis.
4. For the last paragraph of the introduction, the authors switch to past tense, which is at odds with the first part. Perhaps use present tense for the intro, e.g. "Herein, we reconstitute..." while switching to the past tense for the results, if desired. Or else use all past tense. The last sentence of the introduction needs work, perhaps change "on how" to "regarding how."
5. Fig 1/Line 83: In addition to holding proteins above K_d , it could also be important to have their relative concentrations the same as in human cells. Is there a published estimate of each protein's nuclear concentration that could be mentioned here?
6. Figure 1a: suggest to also write role of Lig1 in schematic. Also, consider showing Lig1 as toolbelt model. It would also help the schematic flow to show actual nick translation beyond the RNA primer rather than pointing back to the second cartoon.
7. Fig 1E and 1F should each have a schematic to show which product we are tracking: it is confusing to have both labels present in one schematic.
8. There is minimal discussion about exonuclease activity, yet the kinetics summarized in Fig 1G are

derived from Pol d *exo-*. Line 108 states that control experiments from refs 15 and 26 explain why human Pol d is not faster due to *exo-*, however, a direct discussion about whether the *exo-* mutant modulates the general polymerase speed is lacking. A comment on the difference between the 50 s lanes of WT Pol d and *exo-* Pol d (or any other more direct comparison of kinetics that exists in the literature) is warranted.

9. Fig 1B,D: explain error bars.

10. Fig 3A. Explain how flap cleavage (acceptor departure) is experimentally differentiated from acceptor photobleaching.

11. 4A. A short discussion of PIFQ would be interesting, even though you point to a PIFQ paper from your lab. Particularly, why does the substrate experience PIFQ and not PIFE upon FEN1 binding?

12. Did the authors measure binding of FEN1 binding to a 2nt gap or double flap substrate? If these experiments were excluded, indicate why.

REVIEWER COMMENTS

Reviewer #1 (Remarks to the Author):

In this manuscript, Raducanu and co-workers use a variety of biochemical assays to investigate the dynamics of Okazaki-fragment maturation with purified human proteins. The authors find that this maturation is really quite slow in vitro, and that the processivity of DNA polymerase delta is substantially reduced by competition with FEN1 for nicked or flap substrates. The authors demonstrate that, while Pol delta, FEN1 and DNA ligase can apparently all associate with PCNA on a replication template simultaneously, FEN1 outcompetes Pol delta even after cleaving a flap to regenerate a nick: in turn, FEN1 can be actively removed from a nicked DNA substrate by DNA ligase 1. For lagging-strand substrates with several ribonucleotides at their 5' ends, nick translation must proceed via repeated cycles of alternating Pol delta/FEN1 activity, with this nick translation rendered slow by the preferential binding of FEN1 (as opposed to Pol delta) to nicked substrates. Pre-removal of the RNA primer by RNase H2 leads to faster Okazaki fragment ligation after only a few rounds of nick translation.

Overall, I think this is a very strong piece of work that provides important insights into a fundamental process, and will be of great interest to researchers in the DNA replication field. The data are of high quality, and are clearly presented and described throughout the manuscript: the authors have also been careful not to overstate their findings, and do a nice job discussing potential caveats in the discussion. I strongly recommend publication of this work.

I think that the manuscript is acceptable for publication in its current form, but have a couple of minor suggestions that the authors could address.

We thank the reviewer for the positive feedback and for the subsequent suggestions that significantly improved the manuscript.

We address below the reviewer's comments.

1. The result shown in Figure S1P, namely that Dna2 and FEN1 together inhibit strand-displacement synthesis to a greater extent than FEN1 alone, is very interesting and I would advocate moving it from the supplement to the main figures.

We agree with the reviewer's suggestion, thank you. The panel is now moved to the main figures and its description has been improved in the text to reflect its importance.

2. The data in figure 4 appear to assume that all proteins under investigation remain associated: if this is the case, it would be helpful to clarify and to provide a rationale for this assumption. If such an assumption cannot be made, the authors should state this caveat.

We thank the reviewer for pointing out this aspect. First, loaded PCNA cannot escape from the DNA due to the terminal Biotin-Avidin blocking. Second, FEN1 binds the PCNA-loaded NP with a low-nanomolar affinity of ~5 nM (Figure 5B), while Lig1 also binds the NP with a low-nanomolar affinity of ~3 nM in the absence of PCNA (as shown by Jurkiw et. al. in doi:10.1093/nar/gkaa1297) and probably even lower in the presence of PCNA. In the experiments presented in Figure 4, FEN1 is used at a concentration of 250 nM, while Lig1 is used at a concentration of 500 nM. Under these conditions, both FEN1 and Lig1 should be bound in a proportion of >98%. Equivalently, by ergodicity, each substrate molecule is occupied by the fixed proteins (FEN1 or Lig1) >98% of the time. Moreover, for panel B, Pol δ cannot efficiently release FEN1 from the NP even when employed at much higher concentration (Figure 2D). Similarly, for panel D, FEN1 cannot release Lig1 even when employed at much higher concentration (Figure S4F). Therefore, we assume that in Figure 4B FEN1 is permanently bound forced by concentration and that in Figure 4D Lig1 is permanently bound forced by concentration. We added a statement in the revised manuscript clarifying this important point.

3. The title of the manuscript is a bit overblown: I think something slightly more descriptive and reflective of the results would be appropriate. As a side note, given that this study raises real questions about how human cells accomplish Okazaki fragment processing, one might argue that the work in fact does the opposite of demystify the process.

We agree with the reviewer. We hope that the new title reflects better the findings of the current manuscript.

4. The data presented in the manuscript suggest that, in the absence of substantially different reaction kinetics in vivo, the lagging strand would invariably contain a large amount of DNA synthesized by a proofreading-deficient DNA polymerase (Pol alpha). Although the authors do a nice job highlighting limitations and potential caveats inherent in their biochemical work, the idea that pol alpha contributes significantly to the mature lagging-strand would be hard to reconcile with low mutation rates. I think the authors should address this issue in the discussion.

We agree with the reviewer's observation. We added a paragraph in the discussion of the revised manuscript regarding the removal of the Pol α -synthesized DNA in light of the current results.

Reviewer #2 (Remarks to the Author):

This is an exciting paper about Maturation of Okazaki Fragments (MOF) that addresses some important questions in the field about the enzymatic order-of-operations with precise single-molecule fluorescence experiments. The subject and general findings could ostensibly be suitable for publication in Nature Communications after revision. Generally, the bulk biochemical section is difficult to follow, and needs special attention to make it readable. However, the bulk and single-molecule fluorescence experiments are comprehensively explained, and the interesting conclusions regarding substrate hand-off during MOF are supported. Generally, sufficient information is provided to allow reproduction. The authors converge on a holistic mechanistic model of MOF using evidence from relative affinities, competition assays, and single-molecule kinetics, and structural information. The fact that RNAseH sped of MOF kinetics by such a degree is also of great interest.

The discussion is thorough, the model is nicely presented, and overall, the data in theory would be well received in the field once the following issues are addressed.

We thank the reviewer for the positive feedback and for the subsequent suggestions that significantly improved the manuscript.

We address below the reviewer's comments.

Major Critiques

1. The first section is particularly difficult to follow. In general, please explain the experimental design and the calculations involved explicitly, with careful attention to maximizing reader comprehension.

We thank the reviewer for the suggestion. The experimental design and the calculations are explained in higher detail in the revised manuscript, both in the results and the figure legends.

2. Line 86, an “apparent MOF” is not defined as a variable in the results section. Looking at the figure, a parameter called $T_{\text{obs-12nt RNA}}$ seems to be the same as MOF, but is also not defined (a similar variable needs to be defined in 6D). To try and explain how the data was fit, the figure legend of 1C calls out Eq. 2 of Materials and Methods, but this equation is not relevant. I would have guessed you wanted to point to Eq S15, but neither of these variables are named in your kinetic model either.

We agree with the reviewer on the ambiguity of the parameter names versus the cited equation. In the revised manuscript, we indicate all the exact fitting equations and calculations in the figure legends for improved clarity. The reference to Eq. 2 of Materials and Methods was simply intended to refer to a single-exponential burst kinetics. We also agree with the reviewer that an equation similar to Eq. S15 or more precisely to Eq. S10 could be more appropriate. Nevertheless, we prefer to use the single-exponential burst

equation for simplicity, since Eqs. S15 or S10 refer to a model for single motor moving along the DNA (for example primer extension or at most SD) and do not include the kinetic competition with FEN1 which occurs on every NP and is in fact dominating, or the convolution with the kinetics of ligation. The single-exponential burst, while not making the most out of the data, offers a simple interpretation for its parameter $T_{\text{obs-12nt RNA}}$ as the time needed for the formation of $\sim 100/e=37\%$ of the maximum MOF product amount.

The “Derivation of a fitting model for polymerization kinetics data” section in Supplementary Methods needs an introductory paragraph for lay audiences. Explain more simply at first why the model needed to be developed. If there are differences between your model and other extant models in the literature, please explain those here, along with your rationale for following your methods.

The explanatory paragraph is now added, thank you. The main purpose of the fitting model was to show that the well-established kinetic model used for motors motion along the DNA can be expanded to include the contribution of motor-unbound DNA products while preserving the form of the empirical processivity factor (Eq. S21) and of the macroscopic processivity (Eq. S25). These equations are used for producing Figures S1I and S1J.

3. Line 87: “Since the NT reaction did not proceed considerably beyond the 12nt RNA (Figure 1B), the processing time for RNA removal was estimated to be ~ 13 s/nt.” It’s difficult to understand from this statement where the 13 s/nt estimate for RNA removal was calculated from.

We thank the reviewer for pointing out this aspect. MOF has two main component sub-reactions that occur sequentially: NT for the removal of the RNA region and ligation of the final NP. They occur sequentially since ligation cannot take place before the RNA is completely removed (Figure S1A). The total time needed for MOF is 155.2 s as determined from the single-exponential burst fitting. The ligation step requires 3.3 s. Therefore, the total removal time of the 12nt RNA should be on average = $155.2 - 3.3 = 151.9$ s. Dividing this number by 12nt yields an average RNA processing time of 12.7 s/nt ~ 13 s/nt. This number is only intended to be compared with the number obtained by directly monitoring NT in Figures 1D-F. In the revised manuscript, we further clarified this aspect in the updated figure legend of Figure 1 and in the Results section.

4. In Figure 1H, the mechanism of heparin trapping is unclear. How is heparin specific for DNA-unbound Pol d? Please either perform the experiment using an unlabeled T/P pair as a trap instead or explain why heparin is superior.

We thank the reviewer for pointing out this important aspect. We do not consider that heparin is by default a superior trap as compared to an unlabeled P/T. Nevertheless, in the absence of pre-loaded PCNA, Pol δ binds a P/T junction with very low affinity (in fact we could barely detect any binding up to 500 nM), and therefore such an experiment requires very large amounts of P/T. Heparin was used as a Pol δ trap by a number of

publications (e.g., as shown by Goulian et. al. in doi:10.1093/nar/18.16.4791, as shown by Mondol et. al. in doi:10.1093/nar/gky1321, as shown by Stodola et. al. in doi:10.1038/nsmb.3207). Moreover, the capacity of heparin to trap Pol δ is clearly demonstrated in the results presented in the original data (Figure 1G): under the pre-trap conditions a complete inhibition of the reaction is observed and this is due to the pre-trapping by heparin of Pol δ alone, since all the other proteins are mixed and assembled with the DNA separately.

Nevertheless, the concern by the reviewer prompted us to perform an alternative approach to the heparin trap, by using a polymerase exchange experiment (Figure S1T in the revised manuscript). In this experiment DNA-unbound Pol δ is competed for rebinding by increasing concentrations of catalytically-inactive Pol δ under SD conditions. It can be immediately visualized that in the presence of a high concentration of catalytically-inactive Pol δ , the distribution of the SD products (second lane from the right Figure S1T) is nearly identical to that of the SD products in the presence of the heparin competitor (Figure 1G).

5. It is unclear why the bulk DNA substrates are blocked with neutravidin and is only mentioned in passing. Wouldn't this slow down the kinetics of OF maturation? Or was it for the purposes of comparison to the TIRF experiments? Is it to keep PCNA bound? Please clearly explain rationale.

The main reason for Neutravidin blocking is to prevent PCNA from sliding-off through the free DNA end. This is now clearly stated in the revised manuscript. PCNA sliding-off the free DNA is a much faster process than PCNA re-loading by RFC and therefore, in the absence of Neutravidin blocking the overall PCNA loading level on DNA would be very low. As it can be seen in the original data, in the absence of PCNA both the SD and NT reactions are completely abolished. *In vivo*, PCNA cannot slide-off due to the bilateral ssDNA regions flanking the primer. We also mention that all the SD and NT experiments are performed with excess proteins over DNA and therefore, none of the proteins (including PCNA) need to recycle. While we cannot completely exclude the possibility that Neutravidin blocking biases to some extent the reaction rate, we believe that this interference is minimal mainly due to two arguments. First, with a 28nt primer, Neutravidin is sufficiently far away from the footprint of the NT complex starting from the P/T junction of <23nt (as shown by Lancey et. al. in doi:10.2210/pdb6TNZ/pdb). Second, the reconstitution of the yeast NT system (as shown by Stodola et. al. in doi:10.1038/nsmb.3207) used identical substrate lengths and Neutravidin blocking, yet it supported an ~50-fold higher NT rate, with the yeast complex being nearly identical in size and composition to the Human one.

6. Fig 3C/D. It is unclear in the text why PCNA binding might (or might not) matter. What is the context of this experiment? Also, the mean calculation and/or curve fit are not explained in the legend. It almost appears as though a distinct population arose from PCNA binding in 3C. Could the authors please discuss and demonstrate empirically why this should not be fit to a bimodal distribution?

We agree with the reviewer that the mean calculation was confusing and not explained. We clearly mention here and in the revised manuscript that the presented numbers indicate the raw mean and SEM of the dwell times calculated using raw arithmetic only and not based on histogram fitting. The histograms are for visual purposes only to estimate the data ranges and we prefer not to fit them as we do not have an analytic model for the dwell time distribution. Moreover, we do not draw any conclusion from the histogram shape. In general, we recommend extreme care while fitting such dwell time distributions with either Gamma or multi-exponential models, if the number and nature of sub-step reactions is unknown. There might be multiple rate-limiting sub-steps in FEN1 catalysis and we do not know their number and nature. In other words, even in the absence of PCNA, we prefer not to fit the histogram as we do not currently have an exact kinetic scheme for FEN1 catalytic sub-steps or even at least the number of such sub-steps. Moreover, modality is most often associated with Gaussian mixture models (GMM) which are not appropriate for fitting dwell time distributions. In the same time it is possible that the reviewer is correct and a sub-population appeared due to limited PCNA loading efficiency (~66%, Figure S3E). We mention here and in the revised manuscript that we could not empirically increase this efficiency regardless of the experimental conditions.

In the absence of PCNA, FEN1 cleaves the flap extremely fast (164 ms; Figure 3B) as compared to the NT rate (10900 ms; Figure 1F). The purpose of the single-molecule cleavage experiments is simply to verify if PCNA blocks or sufficiently delays FEN1 cleavage kinetics to the point where FEN1 cleavage becomes comparable with the NT reaction and therefore its limiting step. This is clearly not the case. The average cleavage time increases slightly (210 ms; Figure 3C), yet this ~50 ms increase is irrelevant at the time scale of 10900 ms of the NT reaction. Even if PCNA is loaded only ~66% and a sub-population exists, the entire data range is <550 ms and this should at the very most double if saturation of the sub-population was completed. Therefore, we do not consider that we would gain any additional insight from histogram fitting, since we already know that FEN1 catalysis is not the rate-limiting step that we are looking for pinpointing the slow NT rate. This is rather a control experiment. This was further verified by increasing the number of datapoints included into the histograms. Indeed, no additional datapoints were found beyond >550 ms in the presence of PCNA, despite considerably increasing the sample size.

Minor Critiques

1. The title is vague enough to read like the title of a review. Please consider changing it to directly reflect your findings.

We agree with the reviewer. We hope that the new title reflects better the findings of the current manuscript.

2. Line 21 “They also suggest..”

The phrasing is now corrected, thank you.

3. Line 29, Refs 8 and 9 shouldn't be separated as they both address RNA/DNA synthesis.

The references are now together, thank you.

4. For the last paragraph of the introduction, the authors switch to past tense, which is at odds with the first part. Perhaps use present tense for the intro, e.g. "Herein, we reconstitute..." while switching to the past tense for the results, if desired. Or else use all past tense. The last sentence of the introduction needs work, perhaps change "on how" to "regarding how."

We thank you the reviewer for the suggestions. All the introduction is now switched to present tense and the indicated phrase is improved.

5. Fig 1/Line 83: In addition to holding proteins above K_d , it could also be important to have their relative concentrations the same as in human cells. Is there a published estimate of each protein's nuclear concentration that could be mentioned here?

In the revised manuscript we discuss estimates of the MOF protein concentrations and relative ratios. This indeed was excellent point and we thank the reviewer for guiding us to think in this direction.

6. Figure 1a: suggest to also write role of Lig1 in schematic. Also, consider showing Lig1 as toolbelt model. It would also help the schematic flow to show actual nick translation beyond the RNA primer rather than pointing back to the second cartoon.

We thank the reviewer for pointing this out. The updated schematic incorporates the mentioned points.

7. Fig 1E and 1F should each have a schematic to show which product we are tracking: it is confusing to have both labels present in one schematic.

The updated panels now have a cartoon schematic for each experiment.

8. There is minimal discussion about exonuclease activity, yet the kinetics summarized in Fig 1G are derived from Pol δ exo^- . Line 108 states that control experiments from refs 15 and 26 explain why human Pol δ is not faster due to exo^- , however, a direct discussion about whether the exo^- mutant modulates the general polymerase speed is lacking. A comment on the difference between the 50 s lanes of WT Pol δ and exo^- Pol δ (or any other more direct comparison of kinetics that exists in the literature) is warranted.

We thank the reviewer for pointing out this aspect. Now the revised manuscript includes a direct comparison of the activities of Pol δ WT vs Pol δ Exo^- in both SD and NT (Figures S1K-S). In SD, the speed of the polymerase is indeed modulated by the presence of the exonuclease activity, with the Pol δ WT being ~3-fold slower, probably due to idling and

partitioning from polymerase activity to exonuclease activity before the first NP. In the FEN1-limited NT reaction, the rate of the reaction after the first NP is identical for Pol δ WT vs Pol δ Exo $^-$. This is in line with the observation that FEN1 engages the flap immediately, cleaves it and remains bound to the NP, which also inhibits the slow Pol δ exonuclease activity.

9. Fig 1B,D: explain error bars.

All error bars are explained in the revised manuscript, thank you.

10. Fig 3A. Explain how flap cleavage (acceptor departure) is experimentally differentiated from acceptor photobleaching.

An explanation is provided in the revised manuscript, thank you.

11. 4A. A short discussion of PIFQ would be interesting, even though you point to a PIFQ paper from your lab. Particularly, why does the substrate experience PIFQ and not PIFE upon FEN1 binding?

We added a short discussion of PIFQ in the current manuscript.

12. Did the authors measure binding of FEN1 binding to a 2nt gap or double flap substrate? If these experiments were excluded, indicate why.

FEN1 binds the double flap structure with low nanomolar affinity in the absence of PCNA (as shown by Zaher et. al. in doi:10.1093/nar/gky082) and with sub-nanomolar affinity in the presence of PCNA (as shown by Joudeh in doi:10.25781/KAUST-053H6; unpublished data from our lab). These affinities are much lower than the affinity of Pol δ for a double flap structure, in line with the observation that FEN1 always wins the flap (Figure 2). The affinity constant of FEN1 for the double flap is now mentioned in the revised manuscript. The affinity of FEN1 for a 2nt gap is not included since FEN1 does not seem to compete efficiently with Pol δ gap closure and therefore it is not critical for the findings in the current study. Overall, only on the nick product, their affinity is comparable and relevant for the kinetic competition. For Pol δ all affinities were included since they contribute to understanding the molecular braking mechanism imposed by the block and not only for the comparison with FEN1.

REVIEWERS' COMMENTS

Reviewer #1 (Remarks to the Author):

The authors have responded to all my concerns, and I once again recommend publication of this interesting manuscript

Reviewer #2 (Remarks to the Author):

This manuscript is greatly improved after revision and I strongly recommend it for publication.

Some notes: I may have misunderstood the schematic when I wrote: "It would also help the schematic flow to show actual nick translation beyond the RNA primer rather than pointing back to the second cartoon." I'm not looking at the two in front of me but now it looks unwieldy! I would suggest indicating that several steps are performed more plainly, e.g. bracket the bind/cleave/SD schematic and write "3X" or whatever makes sense. Apologies for the confusion.

For figure 1, I don't think you need to call out the cartoons in D and E, and if you do you could just say (top). As written it's a bit confusing. But including the schematic greatly helps.

The title is improved but you might want to float some other versions to the editor. "remarkably slow" is a little vague, since you really want to say "relatively slow compared to yeast," which wouldn't sound good.

Reviewer #1 (Remarks to the Author):

The authors have responded to all my concerns, and I once again recommend publication of this interesting manuscript.

We thank the reviewer for the positive feedback.

Reviewer #2 (Remarks to the Author):

This manuscript is greatly improved after revision, and I strongly recommend it for publication.

Some notes: I may have misunderstood the schematic when I wrote: "It would also help the schematic flow to show actual nick translation beyond the RNA primer rather than pointing back to the second cartoon." I'm not looking at the two in front of me but now it looks unwieldy! I would suggest indicating that several steps are performed more plainly, e.g. bracket the bind/cleave/SD schematic and write "3X" or whatever makes sense. Apologies for the confusion.

We reverted the panel to the previous version, and we improved the description of the reaction sub-steps.

For figure 1, I don't think you need to call out the cartoons in D and E, and if you do you could just say (top). As written, it's a bit confusing. But including the schematic greatly helps.

We thank the reviewer for this suggestion. The cartoons are not called out separately anymore in the updated manuscript and are referred as top and bottom subpanels.

The title is improved but you might want to float some other versions to the editor. "remarkably slow" is a little vague, since you really want to say "relatively slow compared to yeast," which wouldn't sound good.

We hope that the new title reflects even better the findings of the current study. We agree with the reviewer that slow was intended as a comparison to the yeast system, but it is too long to describe this in the title.